# Learning Hybrid Interpretable Models: Theory, Taxonomy, and Methods

**Julien Ferry**                                                                          *jferry@laas.fr*
*Operations Research, Combinatorial Optimization and Constraints*
*LAAS-CNRS, Université de Toulouse, CNRS*
*Toulouse, France*

**Gabriel Laberge**                                                              *gabriel.laberge@polymtl.ca*
*Génie Informatique et Génie Logiciel*
*Polytechnique Montréal*
*Montréal, Canada*

**Ulrich Aïvodji**                                                               *ulrich.aivodji@etsmtl.ca*
*Software and Information Technology Engineering*
*École de Technologie Supérieure*
*Montréal, Canada*

**Reviewed on OpenReview:** *https://openreview.net/forum?id=XzaSGIStXP*

## Abstract

A Hybrid Interpretable Model involves the cooperation of an interpretable model and a complex black-box. At inference, any input of the model is assigned to either its interpretable or complex component based on a gating mechanism. The ratio of data samples sent to the interpretable component is referred to as the model *transparency*. Despite their high potential, Hybrid Interpretable Models remain under-studied in the interpretability/-explainability literature. In this paper, we remedy this fact by presenting a thorough investigation of such models from three perspectives: Theory, Taxonomy, and Methods. First, we highlight the potential generalization benefits of sending samples to an interpretable component by deriving a Probably-Approximately-Correct (PAC) generalization bound. This guarantee indicates a *sweet spot* for optimal transparency, which suggests that redirecting inputs to an interpretable model can act as regularization. Secondly, we provide a general taxonomy for the different ways of training such models: the *Post-Black-Box* and *Pre-Black-Box* paradigms. These approaches differ in the order in which the interpretable and complex components are trained. We show where the state-of-the-art Hybrid-Rule-Set and Companion-Rule-List fall in this taxonomy. Thirdly, we implement the two paradigms in a single method: HybridCORELS, which extends the CORELS algorithm to Hybrid Interpretable Modeling. By leveraging CORELS, HybridCORELS provides a certificate of optimality of its interpretable component and precise control over transparency. We finally show empirically that HybridCORELS is competitive with existing approaches and performs just as well as a standalone black-box (or even better) while being partly transparent.

## 1 Introduction

The ever-increasing integration of machine learning models in high-stakes decision-making contexts such as healthcare, justice, or finance (*e.g.,* kidney exchange (Aziz et al., 2021), recidivism prediction (Angwin et al., 2016) or credit scoring (Aniceto et al., 2020)) has fostered a growing demand for transparency in recent years. Current workhorses to address transparency concerns in machine learning include black-box explanation and transparent design techniques (Guidotti et al., 2018). Black-box explanation techniques aim at explaining

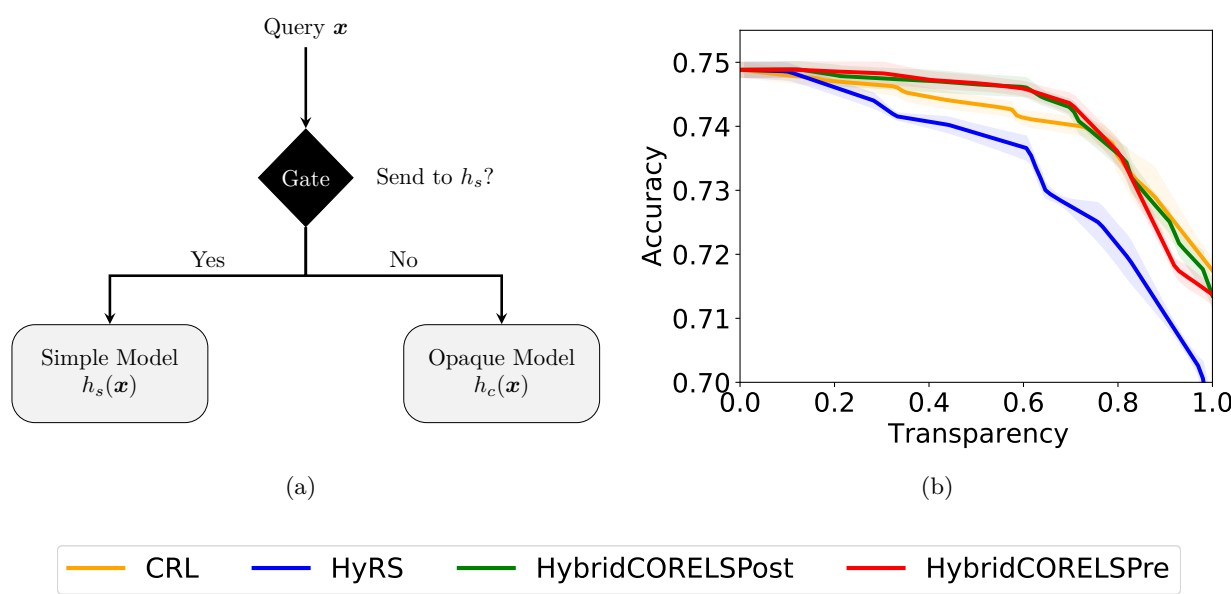

Figure 1: Overview of Hybrid Interpretable Modeling. (a) General schematic of a Hybrid Interpretable Model where, at inference time, a gating mechanism determines whether to send the instance to the interpretable component $h_s$ or to the complex one $h_c$. (b) Letting transparency be the ratio of samples sent to the interpretable component $h_s$, the trade-off between accuracy and transparency can be measured and compared across different Hybrid Interpretable Models.

complex machine learning models in a post-hoc fashion with global explanations such as `Trepan` (Craven & Shavlik, 1995) and `BETA` (Lakkaraju et al., 2017) or local explanations such as `LIME` (Ribeiro et al., 2016) and `SHAP` (Lundberg & Lee, 2017). On the other hand, transparent design concerns the development of inherently interpretable models such as rule lists (Rivest, 1987; Angelino et al., 2017), rule sets (Rijnbeek & Kors, 2010), decision trees (Breiman, 2017), and scoring systems (Ustun & Rudin, 2016).

However, both black-box explanations and transparent design face performance and trustworthiness challenges that can prevent their wide adoption. On the one hand, while inherently interpretable models can be more easily understood and adopted by non-domain experts, their out-of-the-box performance can be worse than non-transparent models. Moreover, training such models to optimality is often NP-hard due to their discrete nature. On the other hand, black boxes can effortlessly attain high performance but their decision mechanisms are opaque and hard to understand by both experts and non-experts. Also, post-hoc explanations of these complex models have been shown to be unreliable and highly manipulable by ill-intentioned entities (Aïvodji et al., 2019; Slack et al., 2020; Dimanov et al., 2020; Aïvodji et al., 2021; Laberge et al., 2022). This conundrum between black-box or transparent designs is colloquially referred to as the "accuracy-transparency trade-off", that is, one has to choose between transparent models with lower performance or opaque models that perform well but whose explanations are not trustworthy. Still, this trade-off is not a quantitative measure but rather a part of the collective imagination of researchers. For this reason, the accuracy-transparency trade-off has been heavily criticized and even labeled a myth (Rudin, 2019). But the question remains, does such a trade-off exist? And if it does, is there a way to quantitatively measure it? Or even optimize it?

To explore such questions, we will not treat black-box and transparent designs as dichotomies. Rather, we will explore the continuum between the two philosophies by studying Hybrid Interpretable Models (Wang, 2019; Pan et al., 2020; Wang & Lin, 2021): predictors that involve the cooperation of an interpretable model and a complex black box. At inference time, any input is assigned to either its interpretable or complex component based on a gating mechanism, see Figure 1 (a). The intuition behind this type of modeling is that not all examples in a dataset are hard to classify.

The model's transparency is defined as the ratio of samples that are sent to the interpretable part (Wang, 2019; Pan et al., 2020; Wang & Lin, 2021). The higher the transparency, the more model predictions can be understood and certified. However, the interpretable component could make more errors than its complex counterpart. Therefore, an integral part of Hybrid Interpretable Models is to empirically explore the accuracy-transparency trade-off and find the best compromises, see Figure 1 (b). Note that the accuracy-transparency trade-off becomes something we measure and optimize.

Despite their high potential, Hybrid Interpretable Models remain under-studied in the interpretability/explainability literature. One of the reasons could be that learning interpretable models is very hard (often NP-Hard), and fitting a Hybrid Interpretable Model on top can only be harder. To address this issue, past studies have optimized such models using simulated annealing heuristics (Wang, 2019; Pan et al., 2020). More specifically, they employed a rule-set/rule-list as the interpretable component and trained it by randomly adding/removing/permuting rules for a fixed number of steps. Nevertheless, we show empirically that the stochasticity of these simulated annealing heuristic hinders the ability of practitioners to consistently attain a target level of transparency. Given the recent development of highly efficient libraries for training interpretable models to optimality (*e.g.*, CORELS for rule-lists (Angelino et al., 2017), GOSDT for decision trees (Hu et al., 2019)), it may now be possible to train Hybrid Interpretable Models to optimality, even when adding a hard constraint on transparency level.

To encourage additional research, we offer a fundamental investigation of such models from three perspectives: Theory, Taxonomy, and Methods, focusing on binary classification for the Theory and Methods parts. From the theory point of view, we explore Probably-Approximately-Correct (PAC) generalization bound of Hybrid Interpretable Models. The aim of this guarantee is not to provide tight bounds directly relevant to practitioners, but rather to highlight the potential generalization benefits of sharing samples between a complex and simple model. This is evidenced by a *sweet spot* of the bound's tightness w.r.t the model transparency. Secondly, we provide a general taxonomy for the different ways of training Hybrid Interpretable Models: the *Post-Black-Box* and *Pre-Black-Box* paradigms. These approaches differ in the order in which the interpretable and complex components are trained. We show where state-of-the-art Hybrid Interpretable Models fall in this taxonomy. Thirdly, we implement the two paradigms in a single method: HybridCORELS, which extends the library CORELS. By leveraging CORELS, HybridCORELS provides a certificate of optimality of its interpretable component and precise control over transparency. We finally show empirically that HybridCORELS is competitive with existing Hybrid Interpretable Models, and performs just as well as a standalone black box (or even better) while being partly transparent. To resume, our contributions are:

- We theoretically study Hybrid Interpretable Models under the PAC-Learning framework and derive a generalization bound. We show that said bound depends on the amount of data classified by each component of the Hybrid Interpretable Model and that an optimal transparency value exists.

- We introduce a taxonomy of Hybrid Interpretable Models' learning methods, identifying two main families: the *Pre-Black-Box* paradigm and the *Post-Black-Box* paradigm. We instantiate the proposed *Pre-Black-Box* paradigm using a notion of *black-box specialization via re-weighting*. In a nutshell, within the *Pre-Black-Box* paradigm, input regions handled by the black-box are known before it is trained. Thus, one can specialize the black-box on said regions by assigning larger weights to the appropriate training samples.

- We review state-of-the-art methods for learning rule-based Hybrid Interpretable Models, and show that they all fall into the *Post-Black-Box* category.

- We extend the CORELS algorithm for learning optimal rule lists into HybridCORELS$_{\text{Post}}$ and HybridCORELS$_{\text{Pre}}$, which learn rule-based Hybrid Interpretable Models within the *Post-Black-Box* and *Pre-Black-Box* paradigms respectively. Both implementations provide optimality guarantees in terms of training accuracy and explicit control of the model *transparency*.

- We empirically compare HybridCORELS$_{\text{Pre}}$ and HybridCORELS$_{\text{Post}}$ with state-of-the-art methods for learning rule-based Hybrid Interpretable Models. Both methods offer competitive trade-offs between *accuracy* and *transparency*.

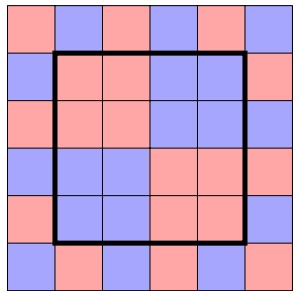

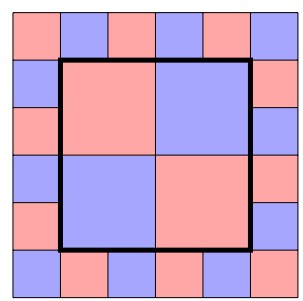

(a) Example of a region $\Omega$ (shown as a thick square) where a complex model $h_c \in \mathcal{H}_c$ (with $|\mathcal{H}_c| = 2^{36}$) is overly complex.

(b) The complex model $h_c$ can be replaced by a simpler one $h_s \in \mathcal{H}_s$ (with $|\mathcal{H}_s| = 2^4$). The Hybrid Interpretable Model has size $|\mathsf{Hyb}| = 2^{24}$.

Figure 2: Toy example with $\mathcal{X} = [0,1] \times [0,1]$. Here the complex models $\mathcal{H}_c$ are all the ways to color the 36 width-1 squares. The simpler models $\mathcal{H}_s$ are all the ways to color the 4 width-2 squares in the middle.

## 2 Hybrid Interpretable Models: a Theoretical Analysis

This section formally introduces Hybrid Interpretable Models and analyzes them under the PAC-Learning framework. A generalization bound is derived and the potential benefit of sharing samples between models is highlighted.

### 2.1 Definitions

Let $\mathcal{X}$ be the input space and let $\mathcal{H}_c, \mathcal{H}_s$ be two sets of binary classifiers $h : \mathcal{X} \to \{0, 1\}$. We shall impose that $|\mathcal{H}_s| < |\mathcal{H}_c| < \infty$ so that $\mathcal{H}_s$ represents a simple set of models while $\mathcal{H}_c$ represents a complex set of models. Finally, we let $\mathcal{P}$ be a set of subsets of $\mathcal{X}$ (for instance, $\mathcal{P}$ may be the power set of $\mathcal{X}$, or the set of linear half-spaces). The intuition behind Hybrid Interpretable Modeling is that there may exist a region $\Omega \in \mathcal{P}$ where a complex model $h_c \in \mathcal{H}_c$ is overkill and could be replaced by a simpler model $h_s \in \mathcal{H}_s$ without significant drop in performance. Formally, a Hybrid Interpretable Model is a triplet $\langle h_c, h_s, \Omega \rangle \in \mathsf{Hyb} := \mathcal{H}_c \times \mathcal{H}_s \times \mathcal{P}$ that instantiates a function of the form

$$\forall \boldsymbol{x} \in \mathcal{X}, \quad \langle h_c, h_s, \Omega \rangle(\boldsymbol{x}) = \begin{cases} h_s(\boldsymbol{x}) & \text{if } \boldsymbol{x} \in \Omega, \\ h_c(\boldsymbol{x}) & \text{otherwise.} \end{cases}$$

Figure 2 presents an informal argument favoring this modeling choice. We will additionally assume that the smaller hypothesis space $\mathcal{H}_s$ involves models that are interpretable such as rule lists, sparse decision trees, scoring systems, etc. This assumption will not affect the theoretical analysis, which will just rely on $|\mathcal{H}_s|$ being small, but it will specify the desiderata of the Hybrid Interpretable Models. Indeed, if $h_s$ is interpretable, then we would like the region $\Omega$ on which it operates to be as big as possible without hindering performance. The size of $\Omega$ is called the *transparency*.

**Definition 1 (Transparency)** *Letting $\mathcal{D}$ be a distribution over $\mathcal{X} \times \{0, 1\}$ representing the binary classification task and $\mathcal{D}_{\mathcal{X}}$ be its marginal over $\mathcal{X}$, the transparency*

$$C_\Omega := \mathop{\mathbb{P}}_{\boldsymbol{x} \sim \mathcal{D}_{\mathcal{X}}} [\boldsymbol{x} \in \Omega] \tag{1}$$

*is the ratio of samples sent to the interpretable component. In opposition, the* opacity $C_{\overline{\Omega}} := \mathbb{P}_{\boldsymbol{x} \sim \mathcal{D}_{\mathcal{X}}}[\boldsymbol{x} \in \overline{\Omega}]$ *is the ratio sent to the black-box.*

The rest of this section is structured as follows: in Section 2.2 we prove that finite Hybrid Interpretable Models (*i.e.,* $|\mathsf{Hyb}| < \infty$) are PAC-Learnable. That is, if we fit them using a finite dataset with sufficiently many examples, then generalization to unseen samples is guaranteed. Afterward, Section 2.3 studies the impact of transparency on the bound, and a "sweet spot" for transparency is highlighted.

## 2.2 PAC-Learnability

The PAC-Learnability framework requires the Realizability Assumption (Shalev-Shwartz & Ben-David, 2014, Definition 2.1).

**Assumption 1 (Realizability)** *There exists a model $\langle h_c^\star, h_s^\star, \Omega^\star \rangle \in$ Hyb that makes perfect predictions on the distribution $\mathcal{D}$ over $\mathcal{X} \times \{0,1\}$:*

$$\mathcal{L}_\mathcal{D}(\langle h_c^\star, h_s^\star, \Omega^\star \rangle) := \underset{(\boldsymbol{x},y)\sim\mathcal{D}}{\mathbb{P}}[\langle h_c^\star, h_s^\star, \Omega^\star \rangle(\boldsymbol{x}) \neq y] = 0. \tag{2}$$

*Intuitively, the predictions of the optimal Hybrid Interpretable Model $\langle h_c^\star, h_s^\star, \Omega^\star \rangle$ match the true label $y$ for any possible input $\boldsymbol{x}$.*

To learn such a model, we can employ the Empirical Risk Minimization (ERM) principle, which consists of sampling a dataset of $M$ iid examples $S := \{(\boldsymbol{x}^{(i)}, y^{(i)})\}_{i=1}^M \sim \mathcal{D}^M$, defining the empirical risk

$$\widehat{\mathcal{L}}_S(\langle h_c, h_s, \Omega \rangle) := \sum_{i=1}^M \mathbb{1}[\langle h_c, h_s, \Omega \rangle(\boldsymbol{x}^{(i)}) \neq y^{(i)}], \tag{3}$$

and minimizing it across Hyb

$$\langle h_c, h_s, \Omega \rangle_S := \text{ERM}_{\text{Hyb}}(S) = \underset{\langle h_c, h_s, \Omega \rangle \in \text{Hyb}}{\arg\min} \widehat{\mathcal{L}}_S(\langle h_c, h_s, \Omega \rangle).$$

Notice that we do not scale the empirical risk by a factor $\frac{1}{M}$ seeing as multiplication by a constant factor does not affect ERM. Solving Equation 2.2 is difficult because it requires fitting two models $h_s, h_c$ as well as the partition over which they operate. Yet, if the optimal partition $\Omega^\star$ is known in advance, it could be treated as fixed so that ERM only focuses on fitting $h_s$ and $h_c$.

**Assumption 2 (Oracle Region)** *Given Hyb and a distribution $\mathcal{D}$ that respect Assumption 1, one has oracle access to the optimal region $\Omega^\star$. Consequently, ERM sets $\Omega = \Omega^\star$ and focuses on fitting $h_s$ and $h_c$.*

**Theorem 2** *Given $|\text{Hyb}| < \infty$ and some $\epsilon > 0$, for any distribution $\mathcal{D}$ where Assumption 1 holds, the following is true for any training set size $M$:*

$$\underset{S\sim\mathcal{D}^M}{\mathbb{P}}[\mathcal{L}_\mathcal{D}(\langle h_c, h_s, \Omega \rangle_S) > \epsilon] \leq \sum_{\Omega\in\mathcal{P}} \mathcal{B}(\epsilon, C_\Omega, \mathcal{H}_c, \mathcal{H}_s, M),$$

*with*

$$\mathcal{B}(\epsilon, C_\Omega, \mathcal{H}_c, \mathcal{H}_s, M) := (1-|\mathcal{H}_c|-|\mathcal{H}_s|e^{-\epsilon M})C_\Omega^M + (1-|\mathcal{H}_s|-|\mathcal{H}_c|e^{-\epsilon M})C_{\overline{\Omega}}^M + |\mathcal{H}_c|(C_{\overline{\Omega}}e^{-\epsilon}+C_\Omega)^M + |\mathcal{H}_s|(C_\Omega e^{-\epsilon}+C_{\overline{\Omega}})^M.$$

*Additionally, if Assumption 2 holds, the bound tightens*

$$\underset{S\sim\mathcal{D}^M}{\mathbb{P}}[\mathcal{L}_\mathcal{D}(\langle h_c, h_s, \Omega \rangle_S) > \epsilon] \leq \mathcal{B}(\epsilon, C_{\Omega^\star}, \mathcal{H}_c, \mathcal{H}_s, M). \tag{4}$$

**Proof** The complete proof is provided in Appendix A. ∎

This generalization bound involves several key quantities: the amount of data $M$, the transparency $C_\Omega$ and opacity $C_{\overline{\Omega}}$ as well as the complexities of the hypothesis spaces $|\mathcal{H}_s|$ and $|\mathcal{H}_c|$. The coming subsection presents how these various parameters impact the bound.

These theoretical bounds have several limitations. First, taking $C_\Omega = 0$ leads to a trivial bound of 1. The same thing occurs when setting $C_\Omega = 1$. Basically, the bound is trivial unless input samples are shared between the complex and simple models. Secondly, the bound requires the knowledge of transparency $C_\Omega = \mathbb{P}_{\boldsymbol{x}\sim\mathcal{D}}[\boldsymbol{x} \in \Omega]$ which cannot be computed exactly in practice since the data-generating distribution $\mathcal{D}$ is unknown. The only way to practically estimate this quantity is to count how many data instances land in the region $\Omega$. Thirdly, the bound can be loose as its computation relies on applying the union bound repeatedly over $\mathcal{P}$, $\mathcal{H}_c$, and $\mathcal{H}_s$. Still, for $C_\Omega \in ]0,1[$, and any $\epsilon \in ]0,1]$ the bound decreases as $M$ increases which implies that learning Hybrid Interpretable Models is possible in theory.

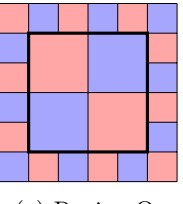 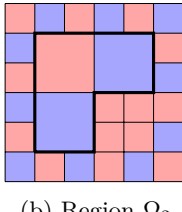 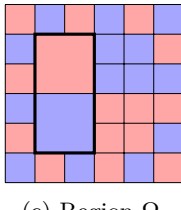 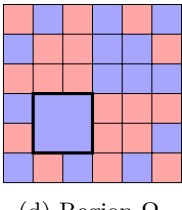

(a) Region $\Omega_1$    (b) Region $\Omega_2$    (c) Region $\Omega_3$    (d) Region $\Omega_4$

Figure 3: Four Hybrid Interpretable Models that are functionally equivalent but have different regions $\Omega$.

## 2.3 Fine-Tuning the Transparency

A particular property of Hybrid Interpretable Models is that the optimal model $\langle h_c^\star, h_s^\star, \Omega^\star \rangle$ from Equation (2) need not be unique. Indeed, given the flexibility of choosing the region $\Omega$ on which the simple model is applied, we could have two models with the same functional output. Figure 3 presents a toy example of four Hybrid Interpretable Models that are all functionally equivalent but with different regions $\Omega$.

Consequently, the Assumption 2 that grants oracle access to $\Omega^\star$ could be generalized to give access to a set of optimal regions $\{\Omega_i\}_{i=1}^R$. Which region should then be returned by the learning algorithm? Using the empirical error as a criterion would not work, since any ERM fitted using these optimal regions would return an error of 0. A more principled approach is to select the region based on the generalization bound. Fixing some region $\Omega_i$ for the ERM algorithm, Equation 4 provides an upper bound $\mathcal{B}(\epsilon, C_{\Omega_i}, \mathcal{H}_c, \mathcal{H}_s, M)$ on the probability of exceeding the error $\epsilon$. To render the analysis independent of the choice of a specific threshold $\epsilon$, we can investigate the Average Bound

$$\overline{\mathcal{B}}(C_\Omega, \mathcal{H}_c, \mathcal{H}_s, M) := \int_0^1 \min\left\{1, \mathcal{B}(\epsilon, C_\Omega, \mathcal{H}_c, \mathcal{H}_s, M)\right\} d\epsilon \tag{5}$$

as a function of the transparency $C_\Omega$. The minimum between 1 and the bound is taken to avoid trivial bounds for probabilities. Note that $\overline{\mathcal{B}}(C_\Omega, \mathcal{H}_c, \mathcal{H}_s, M)$ takes values in $[0, 1]$ and is smaller than 1 if and only if there exists an error $\epsilon$ that occurs with probability $\mathbb{P}_{S \sim \mathcal{D}^M}[\mathcal{L}_\mathcal{D}(\langle h_c, h_s, \Omega \rangle_S) > \epsilon] < 1$.

**Example** Let $\mathcal{H}_s$ as the set of all binary depth-3 decision trees (7 internal nodes and 8 leaves with binary outcomes) fitted on 20 binary features ($\mathcal{X} = \{0, 1\}^{20}$). This hypothesis space has a size $|\mathcal{H}_s| = 2^8 \times 20 \times 19^2 \times 18^4 \approx 1.94 \times 10^{11}$. Let $\mathcal{H}_c$ be any hypothesis space that is larger than $\mathcal{H}_s$ by some factor $|\mathcal{H}_c| = N \times |\mathcal{H}_s|$. Figure 4 (a)&(b) presents the Average Bound as a function of transparency. Given $\mathcal{H}_c$, $\mathcal{H}_s$, and $M$, there is a "sweet spot" where the Average Bound is smallest

$$C_\Omega^\star = \arg\min_{C_\Omega \in [0,1]} \overline{\mathcal{B}}(C_\Omega, \mathcal{H}_c, \mathcal{H}_s, M). \tag{6}$$

Looking more specifically at Figure 4 (a), increasing $N$ reduces the optimal transparency. Simply put, the more complex $\mathcal{H}_c$, the more input samples must be sent to $h_c$ to avoid overfitting. According to Figure 4 (b), the optimal transparency does not seem to vary with $M$, at least when $N = 100$. Figure 4 (c) illustrates the generalization bound as a function of $\epsilon$ when fixing the transparency to its optimal value. We note that the bound is informative since it tells us that the true risk is unlikely to exceed $10\%, 20\%, 40\%$ in various settings.

We conclude this example by emphasizing that Figure 4 is mostly of theoretical interest, so practitioners must take it with a grain of salt. More precisely, the exact values of the "sweet spot" for transparency are not indicative of the values one would obtain in real-life experiments. This is because our analysis is performed on an upper bound, which we hope still captures the generalization dynamics of Hybrid Interpretable Models. In real-life applications, the existence of an optimal transparency must be assessed experimentally.

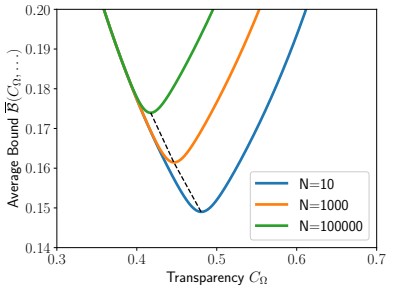
(a) Varying $N$ with $M = 400$.

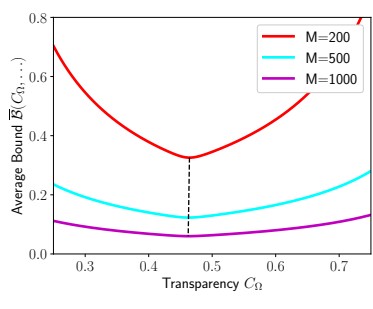
(b) Varying $M$ with $N = 100$.

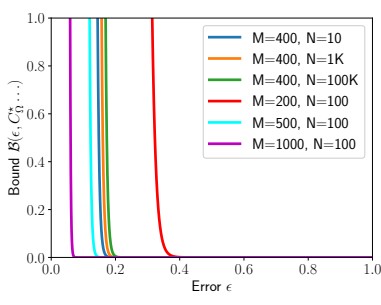
(c) Varying $M$ and $N$.

Figure 4: (a)&(b) Average Bound $\overline{\mathcal{B}}$ as a function of transparency $C_\Omega$. We observe a "sweet spot" with minimal Average Bound, which depends on $N$ (ratio of the hypothesis spaces' sizes $\frac{|\mathcal{H}_c|}{|\mathcal{H}_s|}$). (c) The corresponding generalization bounds when fixing $C_\Omega^\star$ to its optimal value. Note that the bound is informative and tells us that the true risk unlikely exceeds $10\%, 20\%, 40\%$ in various settings.

## 2.4  Takeaways & Improvements

Although the bound makes strong assumptions that may not hold in practical applications, our theoretical analysis leads to fundamental insights:

1. Training Hybrid Interpretable Models is theoretically possible given enough data.

2. Important parameters that influence generalization are the complexities $|\mathcal{H}_s|$ and $|\mathcal{H}_c|$, the transparency $C_\Omega$, and the number of data points $M$.

3. There exists a "sweet spot" of the bound in terms of transparency, suggesting that sharing inputs between a simple and complex model can improve generalization. Later in the manuscript, this will be demonstrated empirically on the COMPAS dataset, see Figure 12 (c).

There are many ways to improve our generalization bound, left for future work.

First, the bound should be extended to account for infinite hypothesis spaces $\mathcal{H}_c$ and $\mathcal{H}_s$ (*e.g.* linear models with few and many coefficients) whose complexity can be characterized with the VC Dimension or Rademacher Complexity (Mohri et al., 2018). This extension is not trivial because bounds based on the VC Dimension employ Sauer's Lemma ((Shalev-Shwartz & Ben-David, 2014, Lemma 6.10), which only holds when $M$ is larger than the VC Dimension. However, bounds for Hybrid Interpretable Models must account for the possibility of sending very few samples to either $h_s$ of $h_c$. We envision leveraging Binomial tail inequalities to bound the probability of these undesirable events.

Second, the Realizability Assumption (Assumption 1) should be removed, allowing for imperfect optimal models $\mathcal{L}_\mathcal{D}(\langle h_c^\star, h_s^\star, \Omega^\star \rangle) > 0$. In that case, the Agnostic-PAC-Learnability framework (Shalev-Shwartz & Ben-David, 2014, Definition 3.3) should be used to bound the probability of failure $\mathbb{P}_{S \sim \mathcal{D}^M}[\mathcal{L}_\mathcal{D}(\langle h_c, h_s, \Omega \rangle_S) > \mathcal{L}_\mathcal{D}(\langle h_c^\star, h_s^\star, \Omega^\star \rangle) + \epsilon]$. Such guarantees would be obtained by leveraging uniform convergence bounds based on the VC Dimension or Rademacher Complexity.

Third, PAC-Bayes guarantees (Alquier et al., 2024) could tighten the union bound over all regions $\sum_{\Omega \in \mathcal{P}}$ by specifying a prior probability distribution over $\mathcal{P}$ favoring high transparency regions. This would allow us to get rid of Assumption 2. Additionally, PAC-Bayes offer an alternative characterization of infinite $\mathcal{H}_c$ and $\mathcal{H}_s$.

# 3 Learning Hybrid Interpretable Models: Taxonomy and Methods

We now introduce our proposed taxonomy of Hybrid Interpretable Models learning frameworks. We then show how rule-based classifiers can be used to implement Hybrid Interpretable Models. Finally, we position state-of-the-art methods within the proposed taxonomy.

## 3.1 Taxonomy of Hybrid Interpretable Models Learning Frameworks

A major challenge in training Hybrid Interpretable Models is that two models must be trained instead of one. Given the proliferation of out-of-the-box implementations of complex model $h_c$, such as `Scikit-Learn` and `XGBoost` classifiers, it would be simpler to rely on them via their pre-existing `fit` and `predict` methods. Henceforth, we encourage Hybrid Interpretable Model training procedures to be *agnostic* to the type of black-box $h_c$.

We now leverage the previous PAC generalization bound to derive a learning objective. As a reminder, we defined in the previous section the data-generating distribution $\mathcal{D}$, the training set $S := \{(\boldsymbol{x}^{(i)}, y^{(i)})\}_{i=1}^M \sim \mathcal{D}^M$, training inputs $S_\mathcal{X} := \{\boldsymbol{x}^{(i)}\}_{i=1}^M$ and the transparency $C_\Omega := \mathbb{P}_{\boldsymbol{x} \sim \mathcal{D}_\mathcal{X}}[\boldsymbol{x} \in \Omega] \approx |S_\mathcal{X} \cap \Omega|/|S_\mathcal{X}|$. It was also demonstrated that two important quantities influencing generalization are the complexity of the simple hypothesis space $\mathcal{H}_s$ and the transparency $C_\Omega$. Since "smaller is better" in any learning objective, we should actually minimize the opacity $C_{\overline{\Omega}} \approx |S_\mathcal{X} \cap \overline{\Omega}|/|S_\mathcal{X}|$. The learning objective would then be

$$\mathsf{obj}(\langle h_c, h_s, \Omega \rangle, S) = \frac{\widehat{\mathcal{L}}_S(\langle h_c, h_s, \Omega \rangle)}{|S|} + \lambda \cdot K_{\mathcal{H}_s} + \beta \cdot \frac{|S_\mathcal{X} \cap \overline{\Omega}|}{|S_\mathcal{X}|}, \tag{7}$$

where $K_{\mathcal{H}_s}$ is a complexity measure of $\mathcal{H}_s$ and $\lambda, \beta \geq 0$ are regularization hyperparameters that control the complexity of $\mathcal{H}_s$ and the opacity $C_{\overline{\Omega}}$. Equation (7) presents the learning of Hybrid Interpretable Models in its most abstract form and we shall make it more specific shortly. We first present several ways to minimize the objective over the space $\mathsf{Hyb} = \mathcal{H}_c \times \mathcal{H}_s \times \mathcal{P}$ that differ on the order in which $h_s$ and $h_c$ are trained.

### 3.1.1 The *Post-Black-Box* Paradigm: Wrapping an Interpretable Model around the Complex One

The *Post-Black-Box* paradigm consists of training the black-box first and then fitting the interpretable model and regions on top. The interpretable components $h_s$ and $\Omega$ can thus be seen as a simplification of $h_c$ in regions where it is overkill. A key advantage of this paradigm is that users owning a pre-trained black-box with high performance can easily wrap an interpretable model on top of it to get an increase of transparency. Furthermore, because it is fitted after the black-box, the interpretable model $h_s$ can be trained with the knowledge of the black-box mistakes, and specifically try to capture the corresponding examples and classify them correctly. In such a case, it effectively corrects these misclassifications from the perspective of the overall Hybrid Interpretable Model. We illustrate the *Post-Black-Box* paradigm in Figure 5 (Top).

### 3.1.2 The *Pre-Black-Box* Paradigm: Black-Box Specialization by Reweighting

In the *Pre-Black-Box* paradigm, the simple model $h_s$ and region $\Omega$ are fitted first, and the black-box is trained on the remaining examples. To avoid overfitting when training the black-box on few samples (since the number of examples left to the black-box can be arbitrarily small), we propose to leverage a weighted training set with higher weights to instances $\boldsymbol{x}^{(i)} \in \overline{\Omega}$ and smaller (but non-zero) weights to instances $\boldsymbol{x}^{(i)} \in \Omega$

$$\forall i \in \{1, 2, \ldots, M\}, \quad w_i = \frac{e^{\alpha \, \mathbb{1}[\boldsymbol{x}^{(i)} \in \overline{\Omega}]}}{\sum_{j=1}^M e^{\alpha \, \mathbb{1}[\boldsymbol{x}^{(j)} \in \overline{\Omega}]}}, \tag{8}$$

The non-uniform weights rely on a **specialization coefficient** $\alpha \geq 0$: the higher $\alpha$, the more $h_c$ focuses on data in $\overline{\Omega}$. The hyperparameter $\alpha$ is fine-tuned in practice. Figure 5 (Bottom) illustrates the *Pre-Black-Box* paradigm pipeline. We note that many classifiers in the `Scikit-Learn` and `XGBoost` packages support non-uniform data weights in their training procedure. Hence, the *Pre-Black-Box* paradigm is also black-box-agnostic.

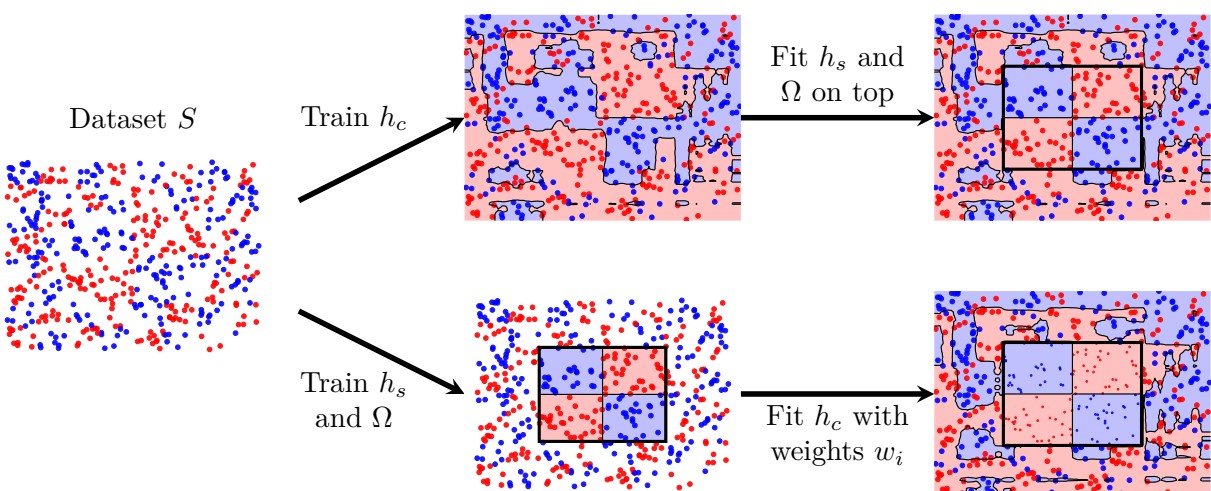

Figure 5: Two paradigms for learning Hybrid Interpretable Models. (Top) In the *Post-Black-Box* paradigm, a black-box is first trained on the whole dataset. Then, the interpretable components are fitted on top of the black-box to simplify it in regions where it is overkill. (Bottom) In the *Pre-Black-Box* paradigm, the interpretable part of the model is trained to identify a region where the task is simple. Afterward, the black-box model is fitted on the data with specialization weights $w_i$ to encourage high performance on instances outside $\Omega$. Here, the weights are visualized as the markers' size.

## 3.2 Rule-Based Modeling

One of the important design choices of a Hybrid Interpretable Model is the space $\mathcal{P}$ of possible subsets $\Omega$ where the interpretable model will operate. An example from previous work is to model these sets via hyperplanes (Wang & Lin, 2021). An alternative is to employ a rule-based model $r$ (*e.g.,* a rule list or a rule set) and define $\Omega_r$ as

$$\Omega_r := \{\boldsymbol{x} \in \mathcal{X} : \mathrm{cover}(r, \boldsymbol{x}) = 1\},$$

where $\mathrm{cover}(r, \boldsymbol{x}) = 1$ if $\boldsymbol{x}$ respects the condition in at least one of the rules in $r$. The advantage of using rule-based models to partition the input space is that they are interpretable by design, hence they can also serve as the simple hypothesis space $\mathcal{H}_s$. That is, we can assign a label to an input depending on which rule captures it.

> **if** $18 \leq \text{Age} \leq 22$ **and** gender=male **then**
>     **return** $y = 1$
> **else if** Prior-Crimes $> 3$ **then**
>     **return** $y = 1$
> **else**
>     **return** $h_c(\boldsymbol{x})$

Since a rule-based model encodes both the region $\Omega$ and the simple function $h_s$ on this region, we can think of rule-based Hybrid Interpretable Models as a tuple $\langle h_c, r \rangle \in \mathcal{H}_c \times \mathcal{H}_s$ instead of a triplet $\langle h_c, h_s, \Omega \rangle$. The learning objective on the training set $S$ becomes

$$\mathrm{obj}(\langle h_c, r \rangle, S) = \frac{\widehat{\mathcal{L}}_S(\langle h_c, r \rangle)}{|S|} + \lambda \cdot |r| + \beta \cdot \frac{|S_{\mathcal{X}} \cap \overline{\Omega}_r|}{|S_{\mathcal{X}}|}, \tag{9}$$

where we measure the complexity of $r$ by its length $|r|$.

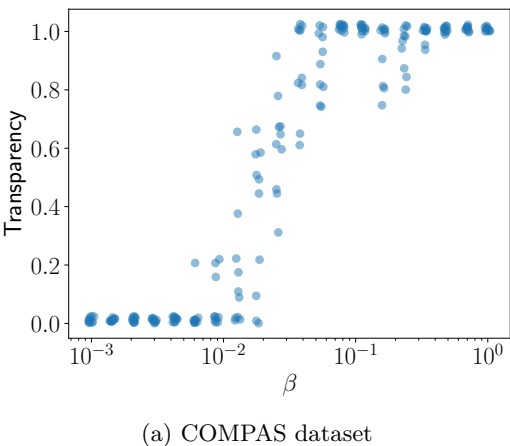

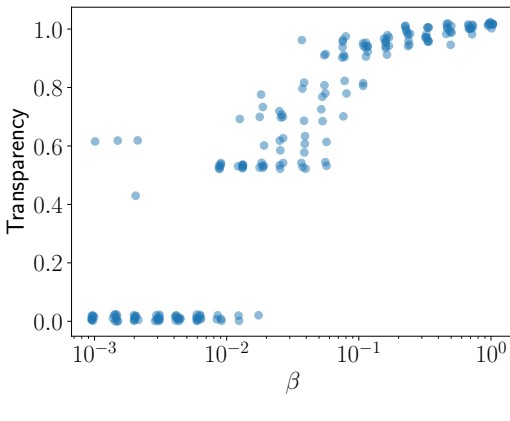

(a) COMPAS dataset

(b) UCI Adult Income dataset

Figure 7: Instability of the transparency of HyRS for different random seeds. A very small jitter was applied to the points to remove juxtapositions.

### 3.3 Rule-Based *Post-Black-Box* Hybrid Interpretable Models

Now that we have introduced several learning paradigms as well as a rule-based Hybrid Interpretable Models, we describe two methods from the literature that apply the *Post-Black-Box* paradigm with rule-sets and rule-lists.

#### 3.3.1 Hybrid Rule-Set (HyRS)

This model has been introduced by Wang (2019) and considers a rule set $r = r_+ \cup r_-$ that combines a set of positive rules $r_+$ and a set of negative rules $r_-$. The resulting tuple $\langle h_c, r \rangle$ takes the form of Figure 6.

The complexity of the interpretable model is the total number of rules $|r|$ and so the learning objective of Equation 9 is used. The minimization of this combinatorial problem is tackled by a simulated annealing search algorithm where neighborhoods are defined as random perturbations of the rule-sets $r_+$ and $r_-$.

**if** cover($r_+, \boldsymbol{x}$) **then**
    **return** $1$
**else if** cover($r_-, \boldsymbol{x}$) **then**
    **return** $0$
**else**
    **return** $h_c(\boldsymbol{x})$

Figure 6: Hybrid Rule-Set.

One of the drawbacks of HyRS is that the user does not have precise control over the transparency $C_\Omega$. There are two design choices in HyRS that lead to this issue. First, the only way to control the desired transparency is to increase the hyperparameter $\beta$ to incentivize larger rule coverage. Still, because a soft constraint is used, one has to conduct a line search over $\beta$ to get the desired transparency. Secondly, since the local search algorithm employed to find the rules is inherently stochastic, several runs of the training procedure with the same hyperparameters can lead to very different models and, by extension, different transparencies. Figure 7 shows different reruns of HyRS on two datasets for 20 different values of $\beta$ that span four orders of magnitude. We see that the relation between transparency and $\beta$ is hardly monotonic because of the variance between reruns. Moreover, the transparency does not vary smoothly w.r.t $\beta$ as seen in the UCI Adult Income dataset, where the transparency jumps from 0 to 0.5 at around $\beta = 10^{-2}$.

#### 3.3.2 Companion Rule-List (CRL)

An alternative method called Companion-Rule-List (CRL) has later been developed in order to address previous limitations (Pan et al., 2020). Notably, CLR streamlines the accuracy-transparency tradeoff by returning multiple Hybrid Interpretable Models with increasing transparency. To see how, given a rule list,

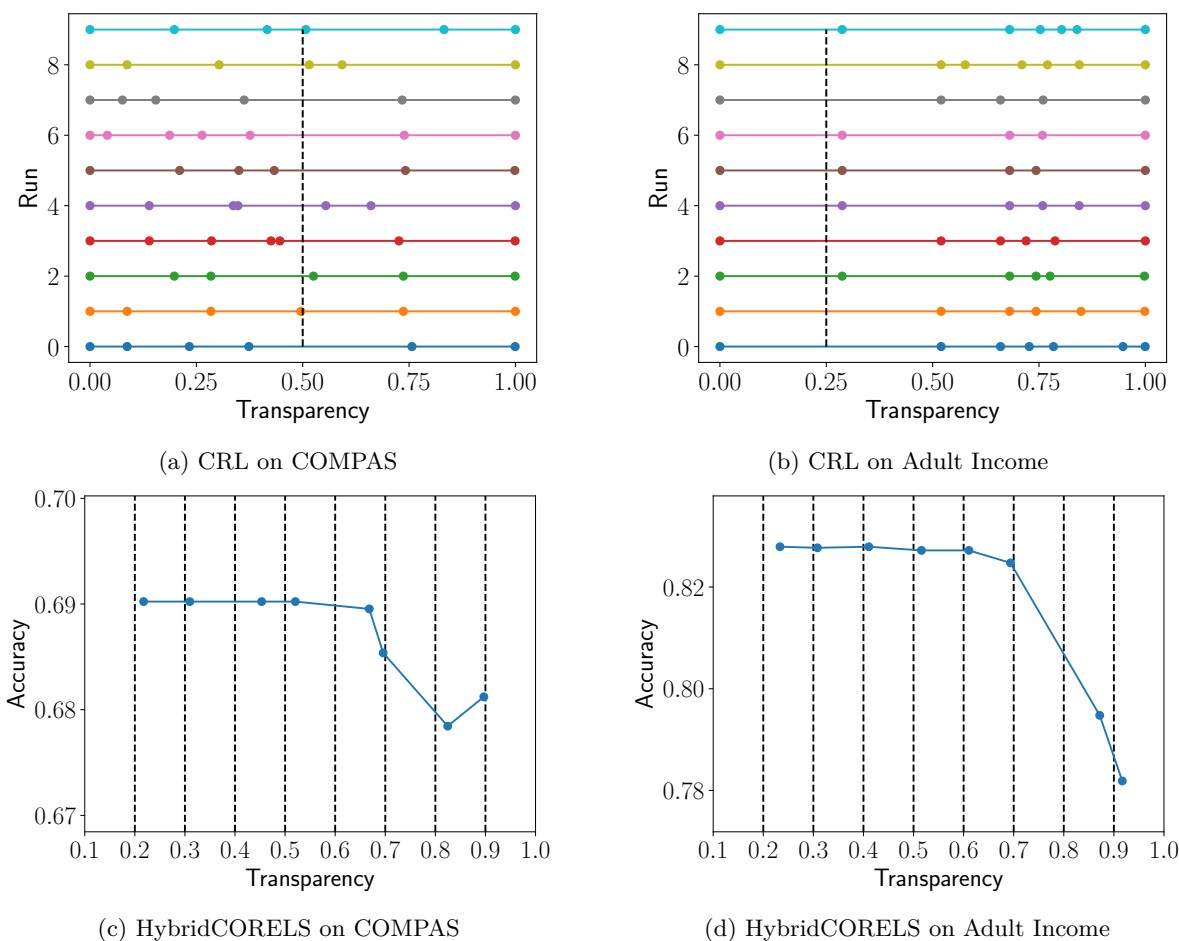

Figure 8: Comparison between the test set transparencies attainable by CRL and HybridCORELS. (a)&(b) The transparencies yielded by CRL runs with various random seeds, each indicated by a different color. (c)&(d) Test set accuracies and transparencies resulting from many runs of HybridCORELS with various transparency constraints $C_{\min} \in [0.2, 0.3, 0.4, 0.5, 0.6, 0.7, 0.8, 0.9]$.

one can insert the black-box at any level of the **else-if** statements. For instance, Figure 9 presents three tuples $\langle h_c, r \rangle$ that are derived from the same list of three rules $r = [r_1, r_2, r_3]$.



**if** cover($r_1, \boldsymbol{x}$) **then**
    **return** 1
**else**
    **return** $h_c(\boldsymbol{x})$

**if** cover($r_1, \boldsymbol{x}$) **then**
    **return** 1
**else if** cover($r_2, \boldsymbol{x}$) **then**
    **return** 0
**else**
    **return** $h_c(\boldsymbol{x})$

**if** cover($r_1, \boldsymbol{x}$) **then**
    **return** 1
**else if** cover($r_2, \boldsymbol{x}$) **then**
    **return** 0
**else if** cover($r_3, \boldsymbol{x}$) **then**
    **return** 1
**else**
    **return** $h_c(\boldsymbol{x})$



Figure 9: A rule list $r = [r_1, r_2, r_3]$ encodes three Hybrid Interpretable Models $\langle h_c, r \rangle$ with increasing transparency (from left to right).

Returning multiple Hybrid Interpretable Models allows users to decide what Hybrid Interpretable Model to use based on their desired transparency. The training objective of CRL is no longer the accuracy but rather the Area-Under-the-Curve (AUC) of the accuracy-transparency curve of the different Hybrid Interpretable Models. A regularization $\lambda \cdot |r|$ is used to avoid long rule-lists. Similarly to HyRS, CRL is trained with

a simulated annealing search algorithm where neighborhoods are defined as random perturbations of the rule-list $r$. Although CRL offers more possibilities for transparency, we find that the inherent stochasticity of the learning procedure still hinders the ability to consistently reach target transparency. Figure 8 (a)&(b) present experiments conducted on the COMPAS and UCI Adult Income where a CRL model was fitted for 10 different random seeds. The different levels of transparency attained by each run are presented as colored lines. For the COMPAS dataset, if a user wishes for a transparency of at least 0.5, then on half of the runs, they would need to go up to about 0.75 transparency. For Adult Income, if an end-user requires transparency of at least 0.25, then on half of the runs, they would need to go up to 0.5 transparency.

The coming section will introduce HybridCORELS, an alternative framework for fitting Hybrid Interpretable Models where desired transparencies are imposed via a hard constraint

$$\frac{|S_{\mathcal{X}} \cap \Omega_r|}{|S_{\mathcal{X}}|} \geq C_{\min} \tag{10}$$

rather than a soft one. As presented in Figure 8 (c)&(d), if a user targets transparency $C_{\min} \in [0.2, 0.3, 0.4, 0.5, 0.6, 0.7, 0.8, 0.9]$, the resulting test set transparency remains close to constraint $C_{\min}$. Unlike Figure 7 (b) and Figure 8 (b), we do not observe a transparency jump from 0 to 0.5 on the Adult Income dataset.

## 4 HybridCORELS: Learning Optimal Hybrid Interpretable Models

We now present our algorithm to learn optimal Hybrid Interpretable Models. First, we introduce the CORELS algorithm, initially proposed to learn optimal rule lists. Then, we describe the integration of a hard constraint on transparency. Finally, we propose HybridCORELS$_{\text{Post}}$ (respectively, HybridCORELS$_{\text{Pre}}$), a modified version of CORELS to learn optimal Hybrid Interpretable Models within the *Post-Black-Box* (respectively, *Pre-Black-Box*) framework.

### 4.1 Learning Optimal Rule Lists: the CORELS Algorithm

Rule lists are interpretable classifiers formed by an ordered list of if-then rules $r$, followed by a default prediction $q_0$ (Rivest, 1987). The set of ordered rules preceding the default prediction is called a prefix. One can observe that any rule list $d = (r, q_0)$ represents a classification function, while any prefix $r$ defines a *partial* classification function, defined within its support $\Omega_r$ (examples matching at least one of the rules within $r$).

To learn Certifiable Optimal RulE ListS, Angelino et al. (2017) proposed CORELS, a branch-and-bound algorithm. It represents the search space of rule lists using a prefix tree, in which each node corresponds to a prefix $r$. Adding a default prediction $q_0$ to $r$ allows the building of a rule list $d = (r, q_0)$. In CORELS' prefix tree, the children nodes of $r$ correspond to prefixes formed by adding exactly one rule at the end of $r$. Thus, the $r$-rooted sub-tree corresponds to all possible extensions of $r$. CORELS' objective function for rule list $d = (r, q_0)$ on dataset $S$ is a weighted sum of classification error and sparsity:

$$\mathsf{obj}(d, S) = \frac{\widehat{\mathcal{L}}_S(d)}{|S|} + \lambda \cdot |r| \tag{11}$$

where $\widehat{\mathcal{L}}_S(d)$ measures the number of errors (incorrect classifications) made by $d$ on $S$ (as defined in (3)), and $|r|$ is the length (number of rules) of rule list $d$'s prefix $r$.

Let $S_r := S \cap (\Omega_r \times \{0, 1\})$ be the subset of $S$ captured by some prefix $r$. Just like any branch-and-bound algorithm, CORELS uses an objective lower bound to prune the prefix tree, and eventually guide the search in a best-first search fashion. For each node of the prefix tree (corresponding to a prefix $r$), it measures the best objective function value that may be reached by extending prefix $r$. If this value is worse than the best solution known so far, then the $r$-rooted sub-tree can be pruned safely. Let $\widehat{\mathcal{L}}_{S_r}(r)$ count the number of mistakes made by prefix $r$ (measured on its support set $S_r$), and $\mathsf{incons}(S)$ denote the minimum number of examples of $S$ that can never be classified correctly, because they have the exact same features vectors as

some other examples, but with a different label. CORELS' objective lower bound for prefix $r$ on dataset $S$ is then computed as follows:

$$\mathsf{lb}(r, S) = \frac{\widehat{\mathcal{L}}_{S_r}(r) + \mathsf{incons}(S \setminus S_r)}{|S|} + (|r| + 1) \cdot \lambda \tag{12}$$

Intuitively, $\widehat{\mathcal{L}}_{S_r}(r) + \mathsf{incons}(S \setminus S_r)$ corresponds to the minimum number of errors that any extension of $r$ can make, given the errors made by $r$ and the errors that can not be avoided due to data inconsistency. CORELS uses several efficient data structures to speed up the computation by breaking down symmetries (Angelino et al., 2017). For instance, a prefix permutation map ensures that only the most accurate permutation of every set of rules is kept. The CORELS pseudo-code is presented as Algorithm 1 in Appendix B.2.

Because it performs global optimization, CORELS builds state-of the-art rule lists in terms of predictive accuracy. On the contrary, exact methods which do not require rules pre-mining do not scale well, and greedy approaches - which are very popular for learning decision trees - usually produce badly performing rule lists. Sections 4.3 and 4.4 will describe how the objective function (11) and its lower bound (12) can be modified to learn Hybrid Interpretable Models within the *Post-Black-Box* and *Pre-Black-Box* paradigms.

### 4.2 Ensuring a User-Defined Transparency Level

As seen in Sections 3.3.1&3.3.2, employing a hard constraint on transparency

$$\frac{|S_r|}{|S|} \geq C_{\min} \tag{13}$$

rather than a soft one (*i.e.,* through a regularization coefficient as done within HyRS/CRL) allows for more precise control over the test set transparency. To enforce constraint (13) using the CORELS branch-and-bound algorithm, we modify the best solution update subroutine, to only perform the update operation if the candidate prefix satisfies the transparency requirement. This guarantees that any returned solution will satisfy (13) while maintaining optimality as the exploration and bounds are not modified.

Even if constraint (13) ensures the strict respect of a user-defined transparency level, we also integrate transparency using a regularization term $\beta \cdot |S \setminus S_r|/|S|$. This allows to break ties: if two models exhibit the same accuracy and sparsity levels, then this regularization term will favor the one with higher transparency. In practice, we set the associated regularization coefficient $\beta$ to a value small enough to only break ties : $\beta < \frac{1}{|S|} \leq \lambda$.

### 4.3 *Post-Black-Box* framework: HybridCORELS$_{\text{Post}}$

We now introduce HybridCORELS$_{\text{Post}}$, a modified version of CORELS producing optimal Hybrid Interpretable Models within the *Post-Black-Box* paradigm. More precisely, HybridCORELS$_{\text{Post}}$ first trains a black-box model (or takes as input a pre-trained black-box model). Then, given a minimum transparency constraint (13), it builds a prefix optimizing the overall model's accuracy and sparsity.

**Objective** Given a black-box $h_c$'s training set predictions, HybridCORELS$_{\text{Post}}$ builds a prefix $r$ capturing at least a proportion of $C_{\min}$ of the training data (transparency constraint (13)), and minimizing the following objective function:

$$\begin{aligned} \mathsf{obj}_{\text{post}}(r, S) &:= \frac{\widehat{\mathcal{L}}_S(\langle h_c, r \rangle)}{|S|} + \lambda \cdot |r| + \beta \cdot \frac{|S \setminus S_r|}{|S|} \\ &= \frac{\widehat{\mathcal{L}}_{S_r}(r) + \widehat{\mathcal{L}}_{S \setminus S_r}(h_c)}{|S|} + \lambda \cdot |r| + \beta \cdot \frac{|S \setminus S_r|}{|S|}. \end{aligned} \tag{14}$$

**Objective lower bound** CORELS' original objective lower bound (12) is still valid and tight in this setup, so we do not need to modify it. Indeed, the error term lower bound $\widehat{\mathcal{L}}_{S_r}(r) + \mathsf{incons}(S \setminus S_r)$ is unchanged, as all remaining black-box errors $\widehat{\mathcal{L}}_{S \setminus S_r}(h_c)$ may potentially be corrected by extending $r$, but the errors already

made by prefix $r$ and those related to remaining inconsistencies can not be avoided. Then, the transparency regularization term can not be used within the objective lower bound, as this term can always reach 0 by sufficiently extending prefix $r$. Finally, the lower bound over the sparsity regularization term still holds: any extension of prefix $r$ must have at least $|r| + 1$ rules.

Finally, HybridCORELS$_{\text{Post}}$ is an exact method: it provably returns a prefix $r$ for which $\mathsf{obj}_{post}(r, S)$ (14) is the smallest among those satisfying the transparency constraint (13). This means that, given fixed black-box predictions and desired transparency level, it produces an optimal Hybrid Interpretable Model in terms of accuracy/sparsity. We provide the HybridCORELS$_{\text{Post}}$ pseudo-code as Algorithm 2 in the Appendix B.3.

### 4.4 *Pre-Black-Box* framework: HybridCORELS$_{\text{Pre}}$

HybridCORELS$_{\text{Pre}}$ is the first algorithm to implement our proposed *Pre-Black-Box* paradigm. It first builds a prefix optimizing accuracy and sparsity, given a minimum transparency constraint (13). Then, it trains the black-box part by specializing it on the uncaptured examples, using the weighting scheme (8). As aforementioned in **Section 3.1.2**, the *Pre-Black-Box* paradigm intrinsically limits the possible collaboration between both parts of tuple $\langle h_c, r \rangle$, as it is not possible for the black-box part to correct the mistakes made by the interpretable part. However, it is possible to consider the inconsistencies left to the black-box part while training the interpretable part, which implements a form of collaboration.

**Objective** HybridCORELS$_{\text{Pre}}$ builds a prefix $r$ capturing at least $C_{\min}$ of the training data (transparency constraint (13)), and minimizing the overall classification error lower bound (based on both prefix $r$'s errors and the inconsistencies let to the black-box part) and sparsity:

$$\mathsf{obj}_{\text{pre}}(r, S) = \frac{\widehat{\mathcal{L}}_{S_r}(r) + \mathsf{incons}(S \setminus S_r)}{|S|} + \lambda \cdot |r| + \beta \cdot \frac{|S \setminus S_r|}{|S|} \tag{15}$$

where the error term $\widehat{\mathcal{L}}_{S_r}(r) + \mathsf{incons}(S \setminus S_r))$ counts the errors of $\langle h_c, r \rangle$ assuming the black-box performs perfectly. It hence provides a tight bound $\widehat{\mathcal{L}}_{S_r}(r) + \mathsf{incons}(S \setminus S_r) \leq \arg\min_{h_c \in \mathcal{H}_c} \widehat{\mathcal{L}}_S(\langle h_c, r \rangle)$.

**Objective lower bound** CORELS' original objective lower bound $\mathsf{lb}(r, S)$ (12) (leveraging both the prefix's errors and the inconsistent examples among the uncaptured ones) is still valid and tight in this setup, so we do not need to modify it. Indeed, the error term is tight: it is not possible for any extension of $r$ to avoid the errors already made by $r$ nor the inconsistencies within the remaining examples. The sparsity term is also tight as any extension of $r$ must have a length of at least $|r| + 1$. As for HybridCORELS$_{\text{Post}}$, the opacity term can not be used within the objective lower bound, as it can always reach 0. An interesting observation is that $\mathsf{lb}(r, S) > \mathsf{obj}_{\text{pre}}(r, S)$ for any prefix $r$ (since $\beta < \lambda$ as indicated in **Section 4.2**). This means that, for any prefix $r$ satisfying the transparency constraint (13)) will not be extended since this can only worsen its objective. So, prefix extensions are only performed in order to meet the transparency constraint (13).

Finally, HybridCORELS$_{\text{Pre}}$ is an exact method: it provably returns a prefix $r$ for which $\mathsf{obj}_{\text{pre}}(r, S)$ (15) is the smallest among those satisfying the transparency constraint (13). This means that, given desired transparency level, it produces an optimal prefix (interpretable part of the final model) in terms of $\langle h_c, r \rangle$ accuracy upper bound and sparsity. If the black-box performs perfectly, then the overall model is certifiably optimal. We provide the HybridCORELS$_{\text{Pre}}$ pseudo-code as Algorithm 3 in the Appendix B.3.

We additionally introduce in the Appendix C another possible implementation of the *Pre-Black-Box* paradigm based on the CORELS algorithm but optimizing an objective function different from that of HybridCORELS$_{\text{Pre}}$. This new variant HybridCORELS$_{\text{Pre,NoCollab}}$ learns a prefix by maximizing its accuracy on the subset $S_r$, without accounting for the task left to the black-box part. Appendix C.1 provides a description of this algorithm and Appendix C.2 empirically compares it with HybridCORELS$_{\text{Pre}}$. The experiments confirm that HybridCORELS$_{\text{Pre,NoCollab}}$ is not competitive with HybridCORELS$_{\text{Pre}}$ in medium to high transparency regimes, due to the lack of collaboration between both parts of the $\langle h_c, r \rangle$ tuple.

# 5 Experiments

In this section, we empirically evaluate our proposed algorithms. We first introduce our experimental setup. Then, we use HybridCORELS$_{\text{Pre}}$ to show that the *Pre-Black-Box* paradigm is suitable to learn Hybrid Interpretable Models, and we report the impact of the specialization coefficient on the performance of the black-box. Afterward, we compare HybridCORELS$_{\text{Pre}}$ and HybridCORELS$_{\text{Post}}$ with two state-of-the-art methods: Hybrid-Rule-Set (HyRS) and Companion-Rule-List (CRL).

## 5.1 Setup

**Datasets** In our experiments, we consider several datasets with various prediction tasks and sizes:

- The **COMPAS** dataset[1] (analyzed by Angwin et al. (2016)) contains 6,150 records from criminal offenders in the Broward County of Florida collected from 2013 and 2014. The corresponding binary classification task is to predict whether a person will re-offend within two years.

- The **UCI Adult Income** dataset (Dua & Graff, 2017) stores demographic attributes of 48,842 individuals from the 1994 U.S. census. Its binary classification task is to predict whether or not a particular person makes more than 50K USD per year.

- The **ACS Employment** dataset (Ding et al., 2021) is an extension of the UCI Adult Income dataset that includes more recent Census data (2014-2018). The goal is to predict if a person is employed/unemployed based on 10 socioeconomic factors. The specific dataset contained information on 203,358 constituents of the Texas state in 2018.

**Rules mining** To ensure a fair comparison between Hybrid Interpretable Models, we pre-mined a set of rules $\Upsilon$ for each dataset. The prefixes were then restricted to select rules $r \in \Upsilon$ so any difference in performance is solely attributable to the learning algorithms and not the quality of the rules. To mine the rules, the datasets were first binarized using quantile for numerical features and one-hot encoding for categorical features. Then, the FP-Growth algorithm (Han et al., 2000) was applied to identify rules of cardinality 1-2 and support of at least 1%. To these sets of rules, we also added the negation of each rule in the original binarized dataset. Finally, the 300 rules with the largest support were kept to generate $\Upsilon$. We ended up with $|\Upsilon| = 230$ rules on COMPAS and $|\Upsilon| = 300$ on the UCI Adult Income and ACS Employment datasets.

**Black-boxes** In all experiments we used the following `Scikit-learn` (Pedregosa et al., 2011) classifiers as black-boxes: a `RandomForestClassifier`, an `AdaBoostClassifier`, and a `GradientBoostingClassifier`. Such black-boxes are in line with the setup considered in the literature (Wang, 2019). We further detail the hyper-parameters tuning of these models in sections 5.2 and 5.3. We note that the Hybrid Interpretable Models studied (HyRS, CRL, and HybridCORELS) are not tied to any specific black-box, nor to a specific implementation. Indeed, they are black-box-agnostic by design.

**Implementation details** Our algorithms HybridCORELS$_{\text{Post}}$ and HybridCORELS$_{\text{Pre}}$ (as well as its HybridCORELS$_{\text{Pre,NoCollab}}$ variant discussed in the Appendix C) are integrated into a user-friendly Python module, publicly available on PyPI[2] and GitHub[3]. They build upon the original CORELS (Angelino et al., 2017) C++ implementation[4] and its Python wrapper[5]. All experiments are run on a computing grid over a set of homogeneous nodes using Intel Platinum 8260 Cascade Lake @2.4Ghz CPU.

**HybridCORELS transparency regularization coefficient $\beta$ setting** In all our experiments using HybridCORELS$_{\text{Pre}}$ or HybridCORELS$_{\text{Post}}$, we set the transparency regularization coefficient $\beta = min(\frac{1}{2 \cdot |S|}, \frac{\lambda}{2})$ to only break ties but ensure that no accuracy nor sparsity will be traded-off for transparency.

---

[1] `https://raw.githubusercontent.com/propublica/compas-analysis/master/compas-scores-two-years.csv`
[2] `https://pypi.org/project/HybridCORELS`
[3] `https://github.com/ferryjul/HybridCORELS`
[4] `https://github.com/corels/corels`
[5] `https://github.com/corels/pycorels`

## 5.2 Exploring the *Pre-Black-Box* Paradigm

**Objective** The objective of this subsection is to assess the appropriateness of the *Pre-Black-Box* paradigm for learning accurate tuples $\langle h_c, r \rangle$. To this end, we use our proposed algorithm implementing this framework: HybridCORELS$_{\text{Pre}}$, depicted in Section 4.4. More precisely, we explore the effect of the *specialization coefficient* on the accuracy of the black-box.

**Setup** For the three datasets presented in Section 5.1, experiments are run for five different train/test splits, with 80% of the data used for training and the remaining 20% for testing. We use HybridCORELS$_{\text{Pre}}$ to produce $\langle h_c, r \rangle$ for several transparency levels: low (0.25), medium (0.5), high (0.75, 0.85) and very high (0.95). For the prefix building part, we optimize the hyperparameters of HybridCORELS$_{\text{Pre}}$ using grid search over the following values: $\lambda \in \{10^{-2}, 10^{-3}, 10^{-4}\}$, $min_{support} \in \{0.01, 0.05, 0.10\}$, and the *objective-guided*, *lower-bound-guided*, and *BFS* search policies. For each experiment, the prefix yielding the best (training) accuracy upper-bound (considering the prefix's errors as well as the inconsistencies left to the black-box part, as depicted in (15)) is retained. The `Scikit-learn` (Pedregosa et al., 2011) black-boxes are chosen to be either an `AdaBoostClassifier` with default parameters, a `GradientBoostingClassifier` with default parameters and a `RandomForestClassifier` with $min\_samples\_split = 10$ and $max\_depth = 10$. The black-boxes are finally trained using different values for the specialization coefficient $\alpha$, ranging from 0 (no specialization) to 10 (highly specialized).

**Results** We first assess the performance of the learned prefixes at various transparency levels by reporting

$$1 - \frac{\widehat{\mathcal{L}}_{S_r}(r) + \text{incons}(S \setminus S_r)}{|S|}, \tag{16}$$

which upper bounds the full model accuracy $1 - \widehat{\mathcal{L}}_S(\langle h_c, r \rangle)/|S|$. Looking at Figure 10, we note that the upper bound decreases for larger transparencies, implying that prefixes with larger support make more errors. Additionally, since prefix transparencies on the training set are very close to the enforced constraint, there must be a conflict between accuracy and transparency. Indeed, if a prefix with very high accuracy and transparency were available, the learning algorithm would systematically select it irrespective of the transparency constraint. We finally observe that transparency generalizes well: the test set transparency levels are very close to the training set ones.

Once the prefixes are learned, we study the effect of the specialization coefficient $\alpha$ on the performances of the downstream black-box. More precisely, for several values of $\alpha$, we report in Figure 11 the train and test set black-box accuracies

$$1 - \frac{\widehat{\mathcal{L}}_{S \setminus S_r}(h_c)}{|S \setminus S_r|} \tag{17}$$

for the `AdaBoostClassifier`. Results for the two other black-boxes are publicly available on our GitHub repository[6]. Note that this accuracy only involves data samples that land outside the support of the prefix, and so it is not representative of the overall accuracy $1 - \widehat{\mathcal{L}}_S(\langle h_c, r \rangle)/|S|$. We focus on three different transparency levels: low (0.25), medium (0.50) and very high (0.95), since the trends observed for high and very high values (0.75, 0.85 and 0.95) are the same.

At a first glance, specialization ($\alpha \geq 1$) appears to improve test black-box accuracy compared to non-specialization ($\alpha = 0$). For ACS-Employ with 0.95 transparency, specialization leads to a 2ppts improvement in test accuracy. Moreover, 0.95 transparency regimes on Adult Income enjoy a 1ppt accuracy increase when setting $\alpha = 1$.

To more systematically assess the benefits of specialization, we define the *improvement rate* as the proportion of runs (out of all our performed experiments) for which a given value of $\alpha \geq 1$ led to an improvement of black-box test accuracy (17) compared to setting $\alpha = 0$. Investigating an improvement rate, rather than an average accuracy over reruns is necessary because different accuracies vary greatly between datasets. Over

---

[6]`https://github.com/ferryjul/HybridCORELS/tree/master/paper/paper_5.2_results.zip`

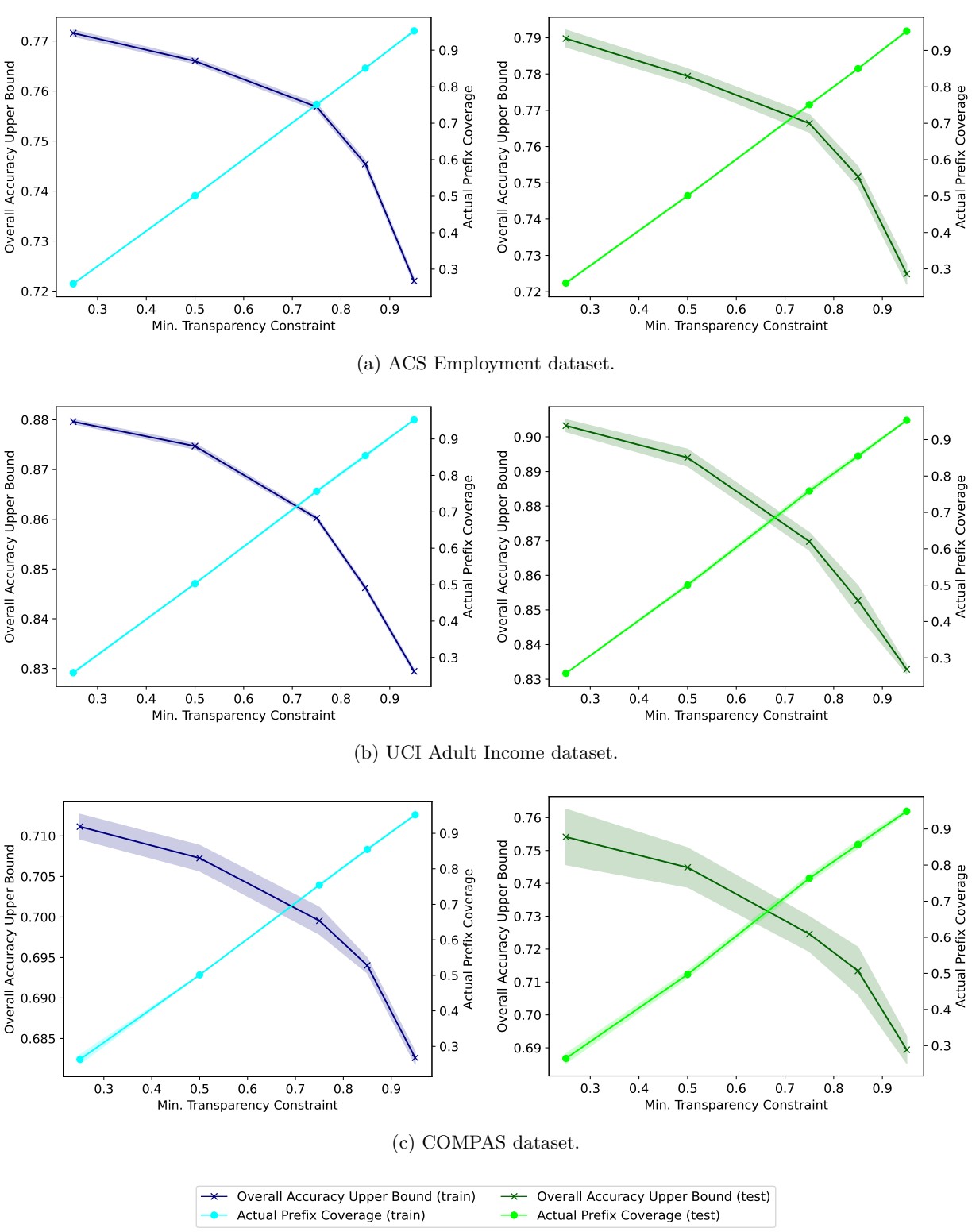

(a) ACS Employment dataset.

(b) UCI Adult Income dataset.

(c) COMPAS dataset.

Figure 10: Training and test performances of the prefixes learned using HybridCORELS$_{\mathrm{Pre}}$. We report the train/test transparencies and accuracy upper bound (Equation 16). The plots show both average values and standard deviation across the five runs with different random seeds.

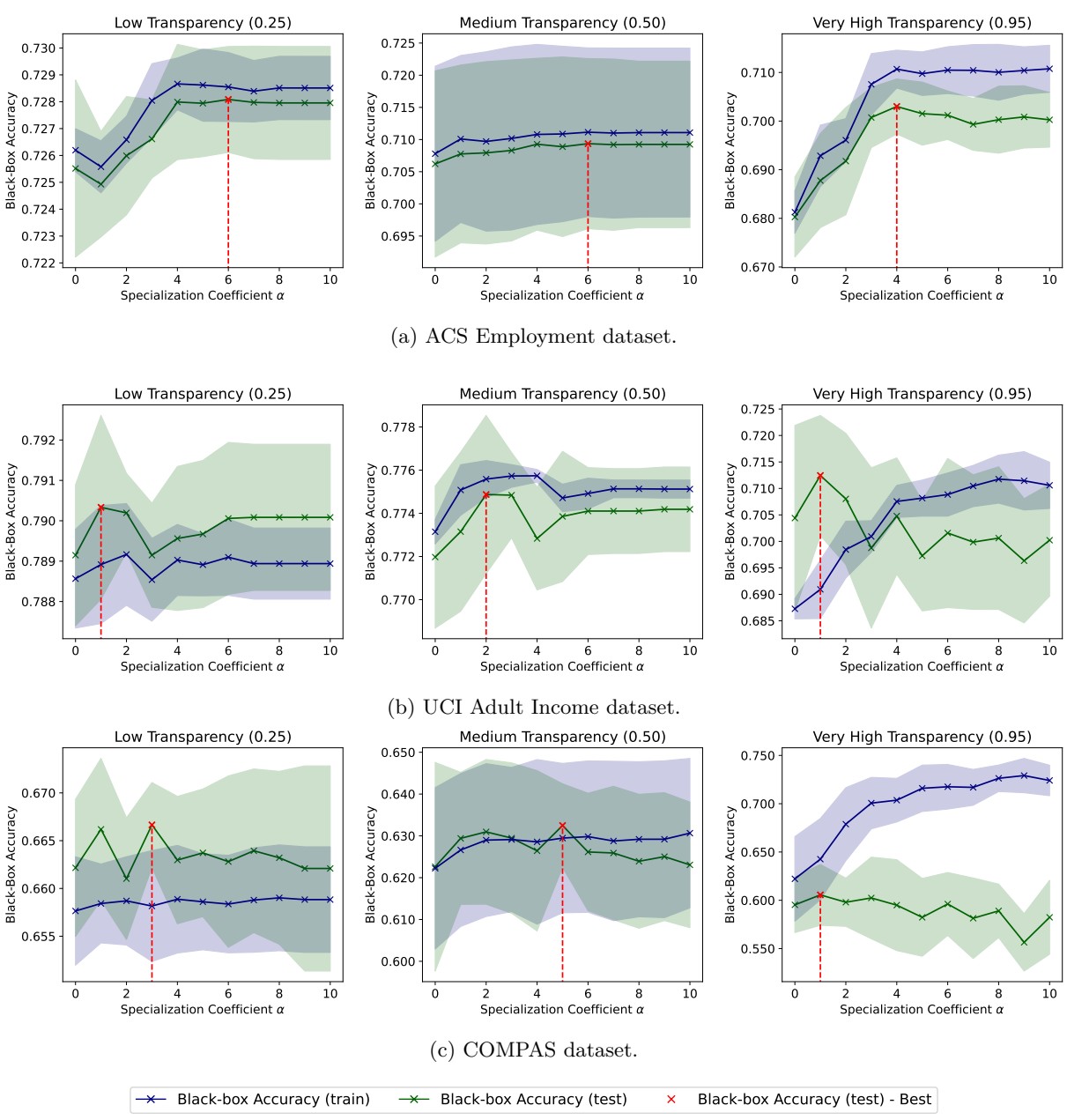

(a) ACS Employment dataset.

(b) UCI Adult Income dataset.

(c) COMPAS dataset.

Figure 11: Training and test accuracy (Equation 17) of the black-box $h_c$ (`AdaBoostClassifier`), learned using HybridCORELS$_{\mathrm{Pre}}$ on different datasets, for different transparency levels. The plots show both average values and standard deviation across the five runs with different random seeds.

all experiments with the `AdaBoostClassifier` black-box, the improvement rate was highest for $\alpha = 2$, with a value of 93.33%. The improvement rate was 73.33% for $\alpha = 5$, and only 66.6% for $\alpha = 8$.

Experiments with the other two black-box types confirm that the improvement rate is higher for moderate specialization $\alpha \in \{1, 2\}$. Consequently, a general "rule of the thumb" is to set $\alpha$ to 1 or 2. For simplicity, the remainder of the manuscript will employ this basic rule. Nevertheless, the optimal specialization is problem dependent, and so future work should investigate tuning it with cross-validation.

We end this section by noting a peculiar pattern: the test set performances presented in Figures 10 & 11 are often better than the training set performance. This might suggest that data leakage is occurring, leading to overoptimistic performance assessment on test. In Appendix D, we lay these concerns to rest by demonstrating that the overall test accuracies $1 - \widehat{\mathcal{L}}_S(\langle h_c, r \rangle)/|S|$ are lower than their training set counterparts. The illusion that test performances are "too good" is induced by the fact that Figures 10 & 11 do not report actual accuracies.

## 5.3 Tradeoffs and Comparison with the State-of-the-Art

**Objective** The aim of this subsection is to explore the trade-offs between the accuracy and transparency of several Hybrid Interpretable Models : the state-of-the-art HyRS and CRL methods, as well as our proposed HybridCORELS$_{\text{Post}}$ and HybridCORELS$_{\text{Pre}}$ algorithms. These experiments serve the secondary purpose of advertising the considerable amounts of transparency that can be attained while maintaining high performance.

**Setup** For these experiments, each dataset was split into training (60%), validation (20%), and test (20%) sets. We randomly generate five such splits and average the results over them. More precisely, for each split, the training set is used to train the models (the prefix and the black-box). The prefix and black-box hyperparameters are optimized using the validation set. Finally, the resulting Hybrid Interpretable Models are evaluated on the test set. Note that while we hereafter focus on such test performances, the training and validation curves show similar trends, and the associated results are all provided on our GitHub repository[7].

We, we detail the training and hyper-parameters optimization procedures for the prefixes and black-boxes.

***Pre-Black-Box* method setup** The experiments using the HybridCORELS$_{\text{Pre}}$ algorithm are divided into two phases. First, for each dataset (out of 3) and each random split (out of 5), we learn prefixes on the training set with 12 different minimum transparency constraints ($C_{\min} \in [0.1, 0.2, 0.3, \ldots, 0.8, 0.9, 0.925, 0.95, 0.975]$). The following hyperparameters values are tried : $\lambda \in \{10^{-2}, 10^{-3}, 10^{-4}\}$, $min_{support} \in \{0.01, 0.05, 0.10\}$, and the *objective-guided*, *lower-bound-guided*, and *BFS* search policies for HybridCORELS$_{\text{Pre}}$. Each prefix learning is limited to a maximum CPU time of 1 hour and a maximum memory use of 8 GB. For each experiment (dataset - random split - minimum transparency), the prefix yielding the best validation accuracy upper bound is kept. In a second phase, for each retained prefix, we try three different `Scikit-learn` (Pedregosa et al., 2011) black-boxes: a `RandomForestClassifier`, an `AdaBoostClassifier`, and a `GradientBoostingClassifier`. The black-box hyperparameters are tuned using the `Hyperopt` (Bergstra et al., 2013) Python library and its Tree of Parzen Estimators (TPE) algorithm, with 100 iterations. Just like the prefixes in the first phase, the black-boxes are trained using the training split (60%) and the hyperparameters are selected based on the validation split (20%) performances. Note that, as for the training set, the validation set loss is weighted to encourage the black-box to accurately classify the examples belonging to $\overline{\Omega}$. Based on the observations from Section 5.2, we adopt a rule of thumb and set the specialization coefficient $\alpha = 1$, which corresponds to a moderate black-box specialization. Optimizing the value of $\alpha$ using the separate validation set (as is done for the other hyperparameters) would be possible, and would likely improve the results, since sticking to one configuration does not allow adapting to the different scenarios. We chose to not do so and let the choice of $\alpha$ as an interesting research avenue, in order to keep the size of the hyperparameters' grid comparable among the different methods.

---

[7]`https://github.com/ferryjul/HybridCORELS/tree/master/paper/results_part_4.zip`

***Post-Black-Box*** **methods setup** Three methods correspond to the *Post-Black-Box* paradigm: HybridCORELS$_{\text{Post}}$, HyRS (Wang, 2019), and CRL (Pan et al., 2020). The experiments using these methods are divided into two phases. First, for each dataset (out of 3) and each random split (out of 5), we train three different `Scikit-learn` (Pedregosa et al., 2011) black-boxes: a `RandomForestClassifier`, an `AdaBoostClassifier`, and a `GradientBoostingClassifier`. The black-box hyperparameters are tuned using the `Hyperopt` (Bergstra et al., 2013) Python library and its Tree of Parzen Estimators (TPE) algorithm, with 100 iterations. The black-boxes are trained using the training split (60%) and their hyperparameters are selected based on the validation split (20%) performances.

In the second phase of the experiments, we train the interpretable parts $\langle h_c, r \rangle$ for the three compared methods, using the $h_c$ learned in the previous phase. The prefix training is performed on the training split (60%), while the hyperparameters values are selected based on the validation split (20%) performances. For HybridCORELS$_{\text{Post}}$, we consider 12 different minimum transparency constraints ($C_{\min} \in [0.1, 0.2, 0.3, \ldots, 0.8, 0.9, 0.925, 0.95, 0.975]$), and the following hyperparameters values: $\lambda \in \{10^{-2}, 10^{-3}, 10^{-4}\}$, $min_{support} \in \{0.01, 0.05, 0.10\}$, and the *objective-guided*, *lower-bound-guided*, and *BFS* search policies. For the HyRS method, similarly to Wang (2019), we use 10 different values for its $\lambda$ hyperparameter (ranging logarithmically from $10^{-3}$ to $10^{-2}$) and 10 different values for its $\beta$ hyperparameter (ranging logarithmically from $10^{-3}$ to $10^0$). For CRL, we consider 10 different values for its *temperature* hyperparameter (ranging linearly from $10^{-3}$ to $10^{-2}$) and 10 different values for its $\lambda$ hyperparameter (ranging logarithmically between $10^{-3}$ and $10^{-1}$). For all three methods HybridCORELS$_{\text{Post}}$, HyRS, and CRL, the hyperparameter grid is roughly of size 100. As in the HybridCORELS$_{\text{Pre}}$ experiments, prefix building is limited to a maximum CPU time of 1 hour and a maximum memory use of 8 GB.

**Final results computation** After trying out all hyperparameters, we are left with a Pareto front representing the Hybrid Interpretable Models that are not dominated in terms of both validations set accuracy and transparency. Still, since the black box and its prefix were fine-tuned on the validation set, we argue that this Pareto front is an over-optimistic description of the true generalization of $\langle h_c, r \rangle$. For this reason, we decided to take the Pareto-optimal models on validation, and compute their accuracy and transparency on the test set, which has not been used yet in this experiment. Hence, we can obtain unbiased measures of the accuracy and transparency for these models. These final measures of accuracy/transparency are used to compare the different approaches and assess whether increasing transparency can lead to equivalent/better generalization.

**Results** The test set accuracy/transparency trade-offs of the different Hybrid Interpretable Models are shown in Figure 12 for each dataset and black-box type. We highlight three main insights from these results.

First, on almost all datasets and black-box types, the methods HybridCORELS$_{\text{Pre}}$ and HybridCORELS$_{\text{Post}}$ are better or equivalent to HyRS and CRL. The only exception is HybridCORELS$_{\text{Pre}}$ in high transparency regimes (0.85-1.0) on the ACS Employment dataset. HybridCORELS offers competitive trade-offs because, given a transparency constraint, it builds the prefix that provably maximizes accuracy, exploring the whole search space of prefixes through a global optimization method. In Figure 13, we show an example tuple $\langle h_c, r \rangle$ for each of the four methods fitted on the same data split (train/validation/test) of the ACS Employment dataset with an AdaBoost black-box. These models were selected on the basis of having the highest test accuracies for a test transparency between 0.6 and 0.8. We note that HybridCORELS$_{\text{Pre}}$ and HybridCORELS$_{\text{Post}}$ are competitive with CRL and even employ similar rules, for example, [`"age_high" and "Female"`], [`"Reference person" and "No disability"`], and [`"age_high" and "Native"`]. HyRS on the other hand, performs worst than the other three since it has a lesser accuracy and transparency.

Second, using HybridCORELS on the ACS Employment and UCI Adult Income datasets, one can reach high transparency values (0.7) while retaining the same performance as the black-box (0.0 transparency). This observation is consistent across all black-box types, which suggests that complex models are often overkill in certain regions of the input space and can safely be replaced by a simpler model on those inputs. From the point of view of certification/maintenance of a machine learning model, being able to assign a majority of inputs to an interpretable component is a tremendous step forward. For instance, since rule lists are interpretable, one might be able to certify that the prefix works properly/safely on the region $\Omega_r$ that

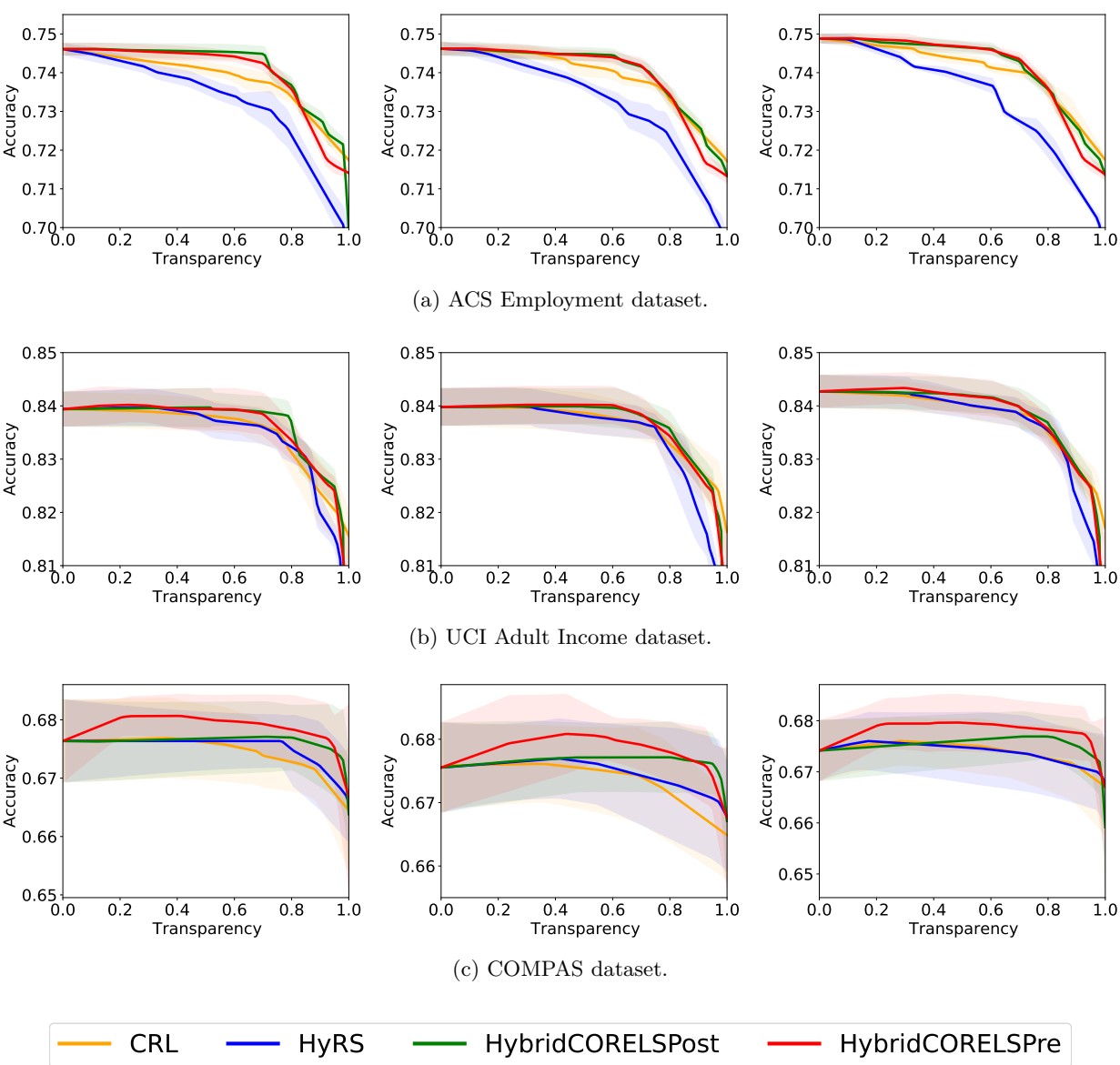

Figure 12: Test set accuracy/transparency trade-offs for various Hybrid Interpretable Models and datasets. The Pareto front for each method is represented as a line and the filled bands encode the std across the five data splits. Results are provided for several black-boxes: (Left) AdaBoost, (Middle) Random Forests, (Right) Gradient Boosted Trees.

```
if ["age_medium" and "No Cognitive difficulty"] then 1
else if ["age_high"] then 0
else
    AdaBoost()
```

(a) HyRS: Test Accuracy 72.8%, Transparency 64.3%

```
if  ["age_high" and "Female"] then 0
else if ["age_high" and "Native"] then 0
else if ["Reference person" and "No disability"] then 1
else if ["Husband/wife" and "No disability"] then 1
else if ["Cognitive difficulty" and "not own child of householder"] then 0
else
    AdaBoost()
```

(b) CRL: Test Accuracy 73.7%, Transparency 75.8%.

```
if ["Disability" and "age_high"] then 0
else if ["Husband/wife" and "Male"] then 1
else if ["age_high" and "Native"] then 0
else if ["age_high" and "Female"] then 0
else if ["Reference person" and "No disability"] then 1
else if ["Bachelor degree"] then 1
else
    AdaBoost()
```

(c) HybridCORELS$_{\text{Pre}}$: Test Accuracy 74.0%, Transparency 70.1%.

```
if ["age_high" and "Female"] then 0
else if ["Husband/wife" and "No disability"] then 1
else if ["age_high" and "Native"] then 0
else if ["Reference person" and "No disability"] then 1
else
    AdaBoost()
```

(d) HybridCORELS$_{\text{Post}}$ : Test Accuracy 73.7%, Transparency 73.0%.

Figure 13: Example of Hybrid Interpretable Models obtained by the different methods on the same data split of the ACS Employment dataset with a AdaBoost black-box.

```
if ["Prior-Crimes=0" and "Age>=30"] then 0
else if ["Prior-Crimes>5" and "Age=24-30"] then 1
else if ["Prior-Crimes=1-3" and "Age>=30"] then 0
else
    RandomForest()
```

(a) HybridCORELS$_{\text{Pre}}$: Test Accuracy 68.1%, Transparency 42.7%

Figure 14: Example Hybrid Interpretable Model obtained by HybridCORELS$_{\text{Pre}}$ on the COMPAS dataset with a Random Forest black-box. Consistent with Figure 12, this model generalizes better than the black-box alone.

will contain the majority of examples seen in deployment. For the minority of instances that fall outside the region, certification might require the verification of the opaque decisions by a committee of domain experts. Such verification might be time-consuming but, the higher the transparency, the fewer examples this committee would need to verify regularly.

Third, when studying HybridCORELS$_{\text{Pre}}$ fitted on COMPAS, one can consistently observe a "sweet spot" for transparency where the generalization is maximal and even better than the standalone black-box. The existence of such a "sweet spot", highlights the regularization effect of sharing inputs between a complex and simple model. This generalization improvement is mainly observed with the HybridCORELS$_{\text{Pre}}$ method,

which constitutes an argument in favor of the *Pre-Black-Box* paradigm. We report in Figure 14 an example model learned with HybridCORELS$_{\text{Pre}}$ on COMPAS, which generalizes better than a standalone black-box. As we observe, just adding three simple rules before the black-box model allows for test accuracy improvements. Note that keeping the same performance while increasing transparency is also desirable (as was the case with the ACS Employment and UCI Adult Income datasets).

## 6    Conclusion

In this paper, we laid the foundations for a promising line of work that was initiated some years ago: hybridizing interpretable and black-box models to "take the best of both worlds" (Wang, 2019; Pan et al., 2020; Wang & Lin, 2021). More precisely, we first provided theoretical evidence that such models have generalization advantages, while also being easier to certify and understand. We then proposed a taxonomy of learning algorithms aimed at producing such models, along with a generic framework implementing the (new) *Pre-Black-Box* paradigm. We introduced algorithms belonging to two identified paradigms, namely *Pre-Black-Box* and *Post-Black-Box*. Compared to state-of-the-art methods, our proposed approaches, coined HybridCORELS$_{\text{Pre}}$ and HybridCORELS$_{\text{Post}}$, certify the optimality of the learned models and provide direct control over the desired transparency level. Our experiments demonstrated the ability of the proposed *Pre-Black-Box* paradigm and the high competitivity of our algorithms with the state-of-the-art. Furthermore, empirical findings suggest that this new paradigm may lead to better-generalizing models. Investigating the reasons for this observation is an interesting future work.

Studying Hybrid Interpretable Models that are not rule-based is a promising research avenue. Going beyond rule lists would require adapting other optimal search-based learning algorithms - for instance those producing optimal sparse decision trees - instead of adapting CORELS as was done in this work. Moreover, the value of the *specialization coefficient* $\alpha$ was set via a "rule of thumb" and optimizing its value with cross-validation on a problem-by-problem basis is an important research direction. Finally, beyond the *Post-Black-Box* and *Pre-Black-Box* frameworks, one could envision a third *end-to-end* paradigm where the black-box and simple models are trained simultaneously. While this approach could provide global optimality guarantees, it is also very challenging because two models must be optimized within a unique framework.

## 7    Broader Impact Statement

Hybrid Interpretable Models explore the accuracy-transparency trade-offs, but they do not address the lack of explainability of black-box models. As a result, Hybrid Interpretable Models are not adequate for critical systems where explanations are required for any model decision. This is because the few decisions relayed to the black-box cannot be faithfully explained. Rather, Hybrid Interpretable Models are potentially useful for tasks where imperfect explanations are *sometimes* acceptable. In such settings, the interpretable component can be used to get the "big picture" of the model behavior, while post-hoc methods (*e.g.* LIME and SHAP) can be used to get imperfect explanations of the black-box on "edge cases".

Moreover, a rule-based model is only as good as its premined rules. A poor collection of rules can cause performance degradation but also discrimination. Indeed, as evidenced by Figure 13, rule-based models can discriminate based on protected attributes such as gender and ethnicity. Consequently, we caution against the premature use of our work and advocate that practitioners exclude protected attributes from their rules before applying HybridCORELS.

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

## A  Proof of Theorem 2

**Theorem 1** *Given* $|\mathsf{Hyb}| < \infty$ *and some* $\epsilon > 0$, *for any distribution* $\mathcal{D}$ *where Assumption 1 holds, the following is true for any training set size* $M$:

$$\mathbb{P}_{S\sim\mathcal{D}^M}[\mathcal{L}_{\mathcal{D}}(\langle h_c, h_s, \Omega\rangle_S) > \epsilon] \leq \sum_{\Omega\in\mathcal{P}} \mathcal{B}(\epsilon, C_{\Omega}, \mathcal{H}_c, \mathcal{H}_s, M),$$

*with*

$$\mathcal{B}(\epsilon, C_{\Omega}, \mathcal{H}_c, \mathcal{H}_s, M) := (1-|\mathcal{H}_c|-|\mathcal{H}_s|e^{-\epsilon M})C_{\Omega}^M + (1-|\mathcal{H}_s|-|\mathcal{H}_c|e^{-\epsilon M})C_{\overline{\Omega}}^M + |\mathcal{H}_c|(C_{\overline{\Omega}}e^{-\epsilon}+C_{\Omega})^M + |\mathcal{H}_s|(C_{\Omega}e^{-\epsilon}+C_{\overline{\Omega}})^M.$$

*Additionally, if Assumption 2 holds, the bound tightens*

$$\mathbb{P}_{S\sim\mathcal{D}^M}[\mathcal{L}_{\mathcal{D}}(\langle h_c, h_s, \Omega\rangle_S) > \epsilon] \leq \mathcal{B}(\epsilon, C_{\Omega^\star}, \mathcal{H}_c, \mathcal{H}_s, M). \tag{18}$$

**Proof**   The distribution $\mathcal{D}$ is fixed apriori and our only assumption about it is that a perfect triplet $\langle h_c^\star, h_s^\star, \Omega^\star\rangle \in \mathsf{Hyb}$ exists. Consequently, we must have $\widehat{\mathcal{L}}_S(\langle h_c, h_s, \Omega\rangle_S) = 0$. Given $\epsilon > 0$, our main objective is to upper bound the probability $\mathbb{P}_{S\sim\mathcal{D}^M}[\mathcal{L}_{\mathcal{D}}(\langle h_c, h_s, \Omega\rangle_S) > \epsilon]$ which corresponds to the probability of "failure" by the ERM model. Letting $\mathsf{Hyb}_\epsilon := \{\langle h_c, h_s, \Omega\rangle \in \mathsf{Hyb} : \mathcal{L}_{\mathcal{D}}(\langle h_c, h_s, \Omega\rangle) > \epsilon\}$ be the set of all "failing" models, we have that

$$\begin{aligned}\mathbb{P}_{S\sim\mathcal{D}^M}[\mathcal{L}_{\mathcal{D}}(\langle h_c, h_s, \Omega\rangle_S) > \epsilon] &\leq \mathbb{P}_{S\sim\mathcal{D}^M}[\exists\langle h_c, h_s, \Omega\rangle \in \mathsf{Hyb}_\epsilon \text{ with } \widehat{\mathcal{L}}_S(\langle h_c, h_s, \Omega\rangle) = 0]\\ &\leq \sum_{\Omega\in\mathcal{P}} \mathbb{P}_{S\sim\mathcal{D}^M}[\exists\langle h_c, h_s, \Omega\rangle \in \mathsf{Hyb}_\epsilon \text{ with } \widehat{\mathcal{L}}_S(\langle h_c, h_s, \Omega\rangle) = 0],\end{aligned} \tag{19}$$

where we have used the union bound over all $\Omega \in \mathcal{P}$. From this point on, we will assume that the domain $\Omega$ is fixed. Consequently, the distribution $\mathcal{D}$ can be seen as a mixture of two distributions $\mathcal{D}_c, \mathcal{D}_s$ with disjoint supports $\overline{\Omega}$ and $\Omega$. Formally, we have $\mathcal{D} = C_{\overline{\Omega}}\mathcal{D}_c + C_{\Omega}\mathcal{D}_s$. The edge cases $\text{supp}(\mathcal{D}) \subset \Omega$ and $\text{supp}(\mathcal{D}) \subset \overline{\Omega}$ are covered by setting $C_{\Omega}=1, C_{\overline{\Omega}}=0$ and $C_{\Omega}=0, C_{\overline{\Omega}}=1$ respectively. Sampling from such a mixture distribution $\mathcal{D}$ is a two-step process. First, one chooses a number of instances $m \sim \text{Bin}(C_{\Omega}, M)$ from a binomial law of $M$ trials and probability $C_{\Omega}$ of success. Then one samples $m$ simple examples $S_s \sim \mathcal{D}_s^m$, and samples $M-m$ hard examples $S_c \sim \mathcal{D}_c^{M-m}$. This leads to

$$\begin{aligned}&\mathbb{P}_{S\sim\mathcal{D}^M}[\exists\langle h_c, h_s, \Omega\rangle \in \mathsf{Hyb}_\epsilon \text{ with } \widehat{\mathcal{L}}_S(\langle h_c, h_s, \Omega\rangle) = 0]\\ &= \mathbb{P}_{\substack{m\sim\text{Bin}(C_{\Omega},M)\\ S_s\sim\mathcal{D}_s^m\\ S_c\sim\mathcal{D}_c^{M-m}}}[\exists\langle h_c, h_s, \Omega\rangle \in \mathsf{Hyb}_\epsilon \text{ with } \widehat{\mathcal{L}}_{S_c\cup S_s}(\langle h_c, h_s, \Omega\rangle) = 0]\\ &= \sum_{m=0}^{M} b(m; C_{\Omega}, M) \mathbb{P}_{\substack{S_s\sim\mathcal{D}_s^m\\ S_c\sim\mathcal{D}_c^{M-m}}}[\exists\langle h_c, h_s, \Omega\rangle \in \mathsf{Hyb}_\epsilon \text{ with } \widehat{\mathcal{L}}_{S_c\cup S_s}(\langle h_c, h_s, \Omega\rangle) = 0],\end{aligned} \tag{20}$$

where $b(m; C_{\Omega}, M) := \binom{M}{m}C_{\Omega}^m(1 - C_{\Omega})^{M-m}$ are the binomial coefficients. This formula has two edges cases $m = 0$ and $m = M$ that occur with probability $C_{\overline{\Omega}}^M$ and $C_{\Omega}^M$. The issue here is that one must bound the population loss of the whole Hybrid Interpretable Model while only one of its sub-models is evaluated on empirical data. This is solved by using trivial bounds

$$\begin{aligned}\mathbb{P}_{S_c\sim\mathcal{D}_c^M}[\exists\langle h_c, h_s, \Omega\rangle \in \mathsf{Hyb}_\epsilon \text{ with } \widehat{\mathcal{L}}_{S_c}(h_c) = 0] &\leq 1\\ \mathbb{P}_{S_s\sim\mathcal{D}_s^M}[\exists\langle h_c, h_s, \Omega\rangle \in \mathsf{Hyb}_\epsilon \text{ with } \widehat{\mathcal{L}}_{S_s}(h_s) = 0] &\leq 1.\end{aligned} \tag{21}$$

Assuming $C_{\Omega} \in ]0,1[$, the probability of these edge cases occurring goes to zero as $M \to \infty$, and the triviality of Equation 21 becomes irrelevant.

**Case $0 < \mathbf{m} < \mathbf{M}$** Since the expected loss can be rewritten

$$\mathcal{L}_{\mathcal{D}}(\langle h_c, h_s, \Omega \rangle) = C_{\overline{\Omega}} \mathcal{L}_{\mathcal{D}_c}(h_c) + C_{\Omega} \mathcal{L}_{\mathcal{D}_s}(h_s),$$

we have that

$$\mathcal{L}_{\mathcal{D}_c}(h_c) \leq \epsilon \text{ and } \mathcal{L}_{\mathcal{D}_s}(h_s) \leq \epsilon \Rightarrow \mathcal{L}_{\mathcal{D}}(\langle h_c, h_s, \Omega \rangle) \leq \epsilon,$$

which implies

$$\langle h_c, h_s, \Omega \rangle \in \mathsf{Hyb}_\epsilon \Rightarrow h_c \in \mathcal{H}_{c,\epsilon} \text{ or } h_s \in \mathcal{H}_{s,\epsilon}, \tag{22}$$

where $\mathcal{H}_{c,\epsilon} := \{h_c \in \mathcal{H}_c : \mathcal{L}_{\mathcal{D}_c}(h_c) > \epsilon\}$ and $\mathcal{H}_{s,\epsilon} := \{h_s \in \mathcal{H}_s : \mathcal{L}_{\mathcal{D}_s}(h_s) > \epsilon\}$ are the sets of complex and simple models "failing" on the distributions $\mathcal{D}_c$ and $\mathcal{D}_s$. Note that the "or" in (22) is not exclusive and both parts of the model may fail simultaneously. Therefore, the following holds

$$\underset{\substack{S_s \sim \mathcal{D}_s^m \\ S_c \sim \mathcal{D}_c^{M-m}}}{\mathbb{P}} [\exists \langle h_c, h_s, \Omega \rangle \in \mathsf{Hyb}_\epsilon \text{ with } \widehat{\mathcal{L}}_{S_c \cup S_s}(\langle h_c, h_s, \Omega \rangle) = 0]$$

$$\leq \underset{\substack{S_s \sim \mathcal{D}_s^m \\ S_c \sim \mathcal{D}_c^{M-m}}}{\mathbb{P}} [\{\exists h_c \in \mathcal{H}_{c,\epsilon} \text{ s.t. } \widehat{\mathcal{L}}_{S_c}(h_c) = 0\} \text{ or } \{\exists h_s \in \mathcal{H}_{s,\epsilon} \text{ s.t. } \widehat{\mathcal{L}}_{S_s}(h_s) = 0\}]$$

$$\leq \underset{S \sim \mathcal{D}_c^{M-m}}{\mathbb{P}} [\exists h_c \in \mathcal{H}_{c,\epsilon} \text{ s.t. } \widehat{\mathcal{L}}_S(h_c) = 0] + \underset{S \sim \mathcal{D}_s^m}{\mathbb{P}} [\exists h_s \in \mathcal{H}_{s,\epsilon} \text{ s.t. } \widehat{\mathcal{L}}_S(h_s) = 0]$$

$$\leq |\mathcal{H}_c| e^{-\epsilon(M-m)} + |\mathcal{H}_s| e^{-\epsilon m},$$

where we have used the inequality $\mathbb{P}_{S \sim \mathcal{D}_s^m}[\exists h_s \in \mathcal{H}_{s,\epsilon} \text{ s.t. } \widehat{\mathcal{L}}_S(h_s) = 0] \leq |\mathcal{H}_s| e^{-\epsilon m}$ (Equation 2.9 of Shalev-Shwartz & Ben-David (2014)), and a similar one for $\mathcal{H}_c$. Going back to Equation (20), we get

$$\underset{S \sim \mathcal{D}^M}{\mathbb{P}} [\exists \langle h_c, h_s, \Omega \rangle \in \mathsf{Hyb}_\epsilon \text{ with } \widehat{\mathcal{L}}_S(\langle h_c, h_s, \Omega \rangle) = 0]$$

$$= \sum_{m=0}^{M} b(m; C_\Omega, M) \underset{\substack{S_s \sim \mathcal{D}_s^m \\ S_c \sim \mathcal{D}_c^{M-m}}}{\mathbb{P}} [\exists \langle h_c, h_s, \Omega \rangle \in \mathsf{Hyb}_\epsilon \text{ with } \widehat{\mathcal{L}}_{S_c \cup S_s}(\langle h_c, h_s, \Omega \rangle) = 0]$$

$$\leq C_{\overline{\Omega}}^M + C_\Omega^M + \sum_{m=1}^{M-1} b(m; C_\Omega, M) \left( |\mathcal{H}_c| e^{-\epsilon(M-m)} + |\mathcal{H}_s| e^{-\epsilon m} \right)$$

$$= C_{\overline{\Omega}}^M + C_\Omega^M + |\mathcal{H}_c| \sum_{m=1}^{M-1} b(m; C_\Omega, M) e^{-\epsilon(M-m)} + |\mathcal{H}_s| \sum_{m=1}^{M-1} b(m; C_\Omega, M) e^{-\epsilon m}$$

$$= C_{\overline{\Omega}}^M + C_\Omega^M + |\mathcal{H}_c| \sum_{m=1}^{M-1} b(m; C_{\overline{\Omega}}, M) e^{-\epsilon m} + |\mathcal{H}_s| \sum_{m=1}^{M-1} b(m; C_\Omega, M) e^{-\epsilon m}$$

$$= (1 - |\mathcal{H}_c| - |\mathcal{H}_s| e^{-\epsilon M}) C_\Omega^M + (1 - |\mathcal{H}_s| - |\mathcal{H}_c| e^{-\epsilon M}) C_{\overline{\Omega}}^M +$$

$$|\mathcal{H}_c| \sum_{m=0}^{M} b(m; C_{\overline{\Omega}}, M) e^{-\epsilon m} + |\mathcal{H}_s| \sum_{m=0}^{M} b(m; C_\Omega, M) e^{-\epsilon m}$$

$$= (1 - |\mathcal{H}_c| - |\mathcal{H}_s| e^{-\epsilon M}) C_\Omega^M + (1 - |\mathcal{H}_s| - |\mathcal{H}_c| e^{-\epsilon M}) C_{\overline{\Omega}}^M +$$

$$|\mathcal{H}_c| (C_{\overline{\Omega}} e^{-\epsilon} + C_\Omega)^M + |\mathcal{H}_s| (C_\Omega e^{-\epsilon} + C_{\overline{\Omega}})^M := \mathcal{B}(\epsilon, C_\Omega, \mathcal{H}_c, \mathcal{H}_s, M).$$

In the last step, the following identity was used identity

$$\sum_{m=0}^{M} b(m; C_\Omega, M) e^{-\epsilon m} = (C_\Omega e^{-\epsilon} + C_{\overline{\Omega}})^M.$$

Finally, combining this with Equation (19) leads to the desired result

$$\underset{S \sim \mathcal{D}^M}{\mathbb{P}} [\mathcal{L}_{\mathcal{D}}(\langle h_c, h_s, \Omega \rangle_S) > \epsilon] \leq \sum_{\Omega \in \mathcal{P}} \mathcal{B}(\epsilon, C_\Omega, \mathcal{H}_c, \mathcal{H}_s, M).$$

Under Assumption 2 stating that the optimal region $\Omega^\star$ is known in advance, the proof is the same except that no union bound over $\Omega \in \mathcal{P}$ is required because ERM is guaranteed to return the region $\Omega^\star$. ∎

# B Pseudo-Codes of the HybridCORELS algorithms

While the CORELS algorithm and our proposed HybridCORELS variants were already introduced in **Section 4**, we describe them in more detail in this appendix section. We first introduce some necessary notation that we later use to provide a detailed pseudo-code and description of the CORELS algorithm. We then depict our proposed variants HybridCORELS$_{\text{Post}}$ and HybridCORELS$_{\text{Pre}}$ for learning Hybrid Interpretable Models.

## B.1 Notations

To formally describe the pseudo-code of the CORELS algorithm and those of our modified HybridCORELS variants, we first need to introduce some more detailed notation. As mentioned in **Section 4.1**, a rule list $d$ consists in an ordered set of rules $r$, called a prefix, followed by a default decision $q_0$. Then, we note: $d = (r, q_0)$. Each individual rule $r_i$ involved within prefix $r$ consists of an *antecedent* $a_i$ ("if" part of the rule, consisting in a Boolean assertion over the features' values) and a consequent $q_i$ ("then" part of the rule, consisting in a prediction). We note: $r_i = a_i \rightarrow q_i$, and $r = (r_1, r_2, \ldots, r_{|r|})$ with $|r|$ the length of prefix $r$.

## B.2 CORELS

The pseudo-code of the CORELS algorithm is provided within Algorithm 1. As mentioned in **Section 4.1**, CORELS is a branch-and-bound algorithm exploring a prefix tree, in which each node corresponds to a prefix $r$ and its children are prefixes formed by extending $r$. At each step of the exploration, the nodes belonging to the exploration frontier are sorted within a priority queue $Q$, ordered according to a given search policy. CORELS implements several such policies, including Breadth First Search, Depth First Search, and several Best First Searches. While these policies define the order in which the nodes of the prefix tree are ordered (and may affect the convergence speed), note that they do not affect optimality, and must all lead to the same optimal objective function value given sufficient time and memory. At each step of the exploration, the most promising prefix $r$ is popped from the priority queue $Q$ (line 4). If its lower bound is greater than the best objective found so far (*i.e.,* $r$ can not lead to a rule list improving the current best objective function), it is discarded. Otherwise, it is used to build a rule list by appending a default prediction $q_0$ (line 6). If the resulting rule list $d$ has a better objective function than the best one reached so far, the current best solution is updated at line 9. Finally, each possible extension of $r$ formed by adding a new rule at the end of $r$ gives a new node which is pushed into the priority queue at line 12. The exploration is completed (and optimality is proved) once the priority queue is empty. Note that efficient data structures are used to cut the prefix tree symmetries: for instance, a prefix permutation map ensures that only the best permutation of every set of rules is kept.

## B.3 HybridCORELS

A key difference between our proposed HybridCORELS algorithms and the original CORELS is that our methods aim at learning prefixes (expressing partial classification functions) while CORELS' purpose is to learn rule lists (classification functions). Both HybridCORELS$_{\text{Post}}$ and HybridCORELS$_{\text{Pre}}$ return prefixes (and not rule lists) and take as input an initial best known prefix $r^0$ satisfying the transparency constraint (while the original CORELS takes as input an initial rule list $d^0$). A simple choice for the initial prefix $r^0$ satisfying the transparency constraint is a constant majority prediction: $r^0 \leftarrow [(True \rightarrow q_0)]$ (whose transparency is 1.0). In practice, we use such trivial initial solution for all our experiments.

The pseudo-code of HybridCORELS$_{\text{Post}}$ is provided in Algorithm 2. Key modifications include the use of a different objective function (14) at line 6, aimed at evaluating the overall Hybrid Interpretable Model's performances. One can note that the computation of the new objective function $\text{obj}_{\text{post}}(r, S)$ requires access

---

**Algorithm 1** CORELS

---

**Input**: Training data $S$ with set of pre-mined antecedents $\Upsilon$; initial best known rule list $d^0$ such that $\mathsf{obj}(d^0, S) = z^0$

**Output**: $(d^*, z^*)$ in which $d^*$ is a rule list with the minimum objective function value $z^*$

1: $(d^c, z^c) \leftarrow (d^0, z^0)$
2: $Q \leftarrow queue(())$           $\triangleright$ Initially the queue contains the empty prefix ()
3: **while** $Q$ not empty **do**           $\triangleright$ Stop when the queue is empty
4:     $r \leftarrow Q.pop()$
5:     **if** $\mathsf{lb}(r, S) < z^c$ **then**
6:        $d \leftarrow (r, q_0)$           $\triangleright$ Set default prediction $q_0$ to minimize training error
7:        $z \leftarrow \mathsf{obj}(d, S) = \frac{\widehat{\mathcal{L}}_S(d)}{|S|} + \lambda \cdot |r|$           $\triangleright$ Compute objective $\mathsf{obj}(d, S)$
8:        **if** $z < z^c$ **then**
9:           $(d^c, z^c) \leftarrow (d, z)$           $\triangleright$ Update best rule list and objective
10:        **for** $a$ in $\Upsilon \setminus \{a_i \mid \exists r_i \in r, r_i = a_i \rightarrow q_i\}$ **do**     $\triangleright$ Antecedent $a$ not involved in $r$
11:           $r_{new} \leftarrow (a \rightarrow q)$           $\triangleright$ Set $a$'s consequent $q$ to minimize training error
12:           $Q.push(r \cup r_{new})$           $\triangleright$ Enqueue extension of $r$ with new rule $r_{new}$
13: $(d^*, z^*) \leftarrow (d^c, z^c)$

---

**Algorithm 2** HybridCORELS$_{\mathrm{Post}}$

---

**Input**: Training data $S$ with set of pre-mined antecedents $\Upsilon$; minimum transparency value $C_{\min}$; initial prefix $r^0$ such that $\frac{|S_{r^0}|}{|S|} \geq C_{\min}$; pre-trained black-box model $h_c$

**Output**: $(r^*, z^*)$ in which $r^*$ is a prefix with the minimum objective function value $z^*$

1: $(r^c, z^c) \leftarrow (r^0, z^0)$
2: $Q \leftarrow queue(())$           $\triangleright$ Initially the queue contains the empty prefix ()
3: **while** $Q$ not empty **do**           $\triangleright$ Stop when the queue is empty
4:     $r \leftarrow Q.pop()$
5:     **if** $\mathsf{lb}(r, S) < z^c$ **then**
6:        $z \leftarrow \frac{\widehat{\mathcal{L}}_{S_r}(r) + \widehat{\mathcal{L}}_{S \setminus S_r}(h_c)}{|S|} + \lambda \cdot |r| + \beta \cdot \frac{|S \setminus S_r|}{|S|}$           $\triangleright$ Compute objective $\mathsf{obj}_{\mathrm{post}}(r, S)$
7:        **if** $z < z^c$ and $\frac{|S_r|}{|S|} \geq C_{\min}$ **then**
8:           $(r^c, z^c) \leftarrow (r, z)$           $\triangleright$ Update best prefix and objective
9:        **for** $a$ in $\Upsilon \setminus \{a_i \mid \exists r_i \in r, r_i = a_i \rightarrow q_i\}$ **do**     $\triangleright$ Antecedent $a$ not involved in $r$
10:           $r_{new} \leftarrow (a \rightarrow q)$           $\triangleright$ Set $a$'s consequent $q$ to minimize training error
11:           $Q.push(r \cup r_{new})$           $\triangleright$ Enqueue extension of $r$ with new rule $r_{new}$
12: $(r^*, z^*) \leftarrow (r^c, z^c)$

---

to the pre-trained black-box $h_c$, which is part of the algorithm's inputs. The original CORELS' lower bound is valid and tight for our new objective (as discussed in **Section 4.3**) so we keep this computation unchanged at line 5. Finally, to ensure that the built prefix satisfies a given transparency constraint (13), this condition is verified at line 7 before updating the current best solution at line 8.

The pseudo-code of HybridCORELS$_{\mathrm{Pre}}$ is provided in Algorithm 3. Again, the objective function computation is modified at line 6 to use our proposed $\mathsf{obj}_{\mathrm{pre}}(r, S)$ objective (15). As before, the original lower bound is still valid (as discussed in **Section 4.4**) so we leave it unchanged at line 5. Just like for HybridCORELS$_{\mathrm{Post}}$, the transparency constraint (13) is checked line 7, right before the current best solution update (line 8). Once the optimal prefix $r^*$ is known, the black-box part can be trained (which is not represented in the pseudo-code) using our proposed specialization scheme as described in **Section 3.1.2**.

Finally, both our proposed approaches are *anytime*: the user can specify any desired running time and memory limits, and the algorithm returns the current best solution and objective value $(r^c, z^c)$ if one of the

---

**Algorithm 3** HybridCORELS$_{\text{Pre}}$

---

**Input**: Training data $S$ with set of pre-mined antecedents $\Upsilon$; minimum transparency value $C_{\min}$; initial prefix $r^0$ such that $\frac{|S_{r^0}|}{|S|} \geq C_{\min}$

**Output**: $(r^*, z^*)$ in which $r^*$ is a prefix with the minimum objective function value $z^*$

1: $(r^c, z^c) \leftarrow (r^0, z^0)$
2: $Q \leftarrow queue(())$          $\triangleright$ Initially the queue contains the empty prefix ()
3: **while** $Q$ not empty **do**          $\triangleright$ Stop when the queue is empty
4:     $r \leftarrow Q.pop()$
5:     **if** $\mathsf{lb}(r, S) < z^c$ **then**
6:        $z \leftarrow \frac{\widehat{\mathcal{L}}_{S_r}(r) + \mathsf{incons}(S \setminus S_r)}{|S|} + \lambda \cdot |r| + \beta \cdot \frac{|S \setminus S_r|}{|S|}$        $\triangleright$ Compute objective $\mathsf{obj}_{\text{pre}}(r, S)$
7:        **if** $z < z^c$ and $\frac{|S_r|}{|S|} \geq C_{\min}$ **then**
8:           $(r^c, z^c) \leftarrow (r, z)$        $\triangleright$ Update best prefix and objective
9:           **for** $a$ in $\Upsilon \setminus \{a_i \mid \exists r_i \in r, r_i = a_i \rightarrow q_i\}$ **do**        $\triangleright$ Antecedent $a$ not involved in $r$
10:              $r_{new} \leftarrow (a \rightarrow q)$        $\triangleright$ Set $a$'s consequent $q$ to minimize training error
11:              $Q.push(r \cup r_{new})$        $\triangleright$ Enqueue extension of $r$ with new rule $r_{new}$
12: $(r^*, z^*) \leftarrow (r^c, z^c)$

---

limits is hit and the priority queue is not empty. Even if optimality is not guaranteed in such case, the ability to precisely bound running times and memory footprints is a very practical feature for real-life applications.

## C    Another *Pre-Black-Box* Implementation for HybridCORELS

In this appendix section, we describe another possible implementation of the *Pre-Black-Box* paradigm based on the CORELS algorithm but optimizing a different objective function. We discuss the theoretical differences with the HybridCORELS$_{\text{Pre}}$ algorithm introduced in **Section 4.4** and empirically compare the two methods.

### C.1    HybridCORELS$_{\text{Pre,NoCollab}}$: Theoretical Presentation

We now introduce another possible variant of CORELS implementing the *Pre-Black-Box* paradigm. We coin it HybridCORELS$_{\text{Pre,NoCollab}}$, because contrary to the HybridCORELS$_{\text{Pre}}$ algorithm introduced in **Section 4.4**, the prefix learning phase of HybridCORELS$_{\text{Pre,NoCollab}}$ does not account for the task left to the black-box part. Instead, the prefix is learned to maximize its own accuracy, which results in the remaining examples (that will be handled by the black-box model) being the hardest ones to classify. While black-box specialization could be helpful to deal with such difficult tasks, we observe that, in practice, it has to deal with many inconsistent examples, which considerably lowers its performances.

**Objective**    HybridCORELS$_{\text{Pre,NoCollab}}$ builds a prefix $r$ capturing at least a proportion of $C_{\min}$ of the training data (transparency constraint (13)), and minimizing the weighted sum of $r$'s classification error and sparsity:

$$\mathsf{obj}_{\text{pre,nocollab}}(r, S) = \frac{\widehat{\mathcal{L}}_{S_r}(r)}{|S_r|} + \lambda \cdot |r| + \beta \cdot \frac{|S \setminus S_r|}{|S|} \tag{23}$$

**Objective lower bound**    CORELS' original lower bound (12) does not hold for objective function (23). Indeed, the difficulty here is that $\mathsf{obj}_{\text{pre,nocollab}}$ quantifies a prefix's error only on the subset of examples that it classifies ($S_r$), hence it is not possible to directly consider the inconsistent examples $\mathsf{incons}(S \setminus S_r)$ as in $\mathsf{lb}$ (12): an extension of $r$ may not capture them at all. To obtain a tight lower bound $\mathsf{lb}_{\text{pre,nocollab}}$, one needs to consider simultaneously the support $S_r$ and errors $\widehat{\mathcal{L}}_{S_r}(r)$ of prefix $r$, as well as the labels cardinalities among each group of inconsistent examples (also called *set of equivalent points* in the context of CORELS (Angelino et al., 2017)). A pre-processing step computes a list $\mathcal{G}$ of *inconsistent groups of examples*.

Each group $g \in \mathcal{G}, g \subset S$ is defined by its number of minority examples $min_g$ (those with the least frequent label among group $g$), and its number of majority examples $maj_g$. In fact, our previously introduced count of unavoidable errors uses such groups for its computation: $\mathsf{incons}(S) = \sum_{g \in \mathcal{G}} (min_g)$. For each group $g \in \mathcal{G}$ not captured by prefix $r$ ($g \not\subset S_r$), we verify whether capturing its examples could lower the current prefix's error rate: $c_{p,g} = \mathbb{1}\left[ \frac{min_g}{min_g + maj_g} \leq \frac{\widehat{\mathcal{L}}_{S_r}(r)}{|S_r|} \right]$. Then:

$$
\mathsf{lb}_{\mathrm{pre,nocollab}}(r, S) = \frac{\widehat{\mathcal{L}}_{S_r}(r) + \sum_{g \in \mathcal{G}, g \not\subset S_r} c_{p,g} \cdot min_g}{|S_r| + \sum_{g \in \mathcal{G}, g \not\subset S_r} c_{p,g} \cdot (min_g + maj_g)}
$$
$$
+ (K_r + 1) \cdot \lambda
\tag{24}
$$

Finally, $\mathsf{lb}_{\mathrm{pre,nocollab}}$ precisely quantifies the best objective function that can be reached based on prefix $r$, by only capturing inconsistent groups of examples that improve the objective function (lowering the error rate). The definition of $c_{p,g}$ uses a less or equal operator because in case the error rate is unchanged after capturing an additional group of inconsistent examples, the operation should be performed as it would increase the coverage (and the associated regularisation term). There exists a (partial) classification function whose error rate is exactly the one computed in $\mathsf{lb}_{\mathrm{pre,nocollab}}$, so this bound is tight.

Finally, HybridCORELS$_{\mathrm{Pre,NoCollab}}$ is an exact method: it provably returns a prefix $r$ for which $\mathsf{obj}_{\mathrm{pre,nocollab}}(r, S)$ (23) is the smallest among those satisfying the transparency constraint (13). This means that, given desired transparency level, it produces an optimal prefix (interpretable part of the final model) in terms of accuracy/sparsity. The pseudo-code of HybridCORELS$_{\mathrm{Pre,NoCollab}}$ is similar to that of HybridCORELS$_{\mathrm{Pre}}$ presented in Algorithm 3, except that the objective function $\mathsf{obj}_{\mathrm{pre}}(r, S)$ and lower bound $\mathsf{lb}(r, S)$ on lines 6 and 5 are replaced by $\mathsf{obj}_{\mathrm{pre,nocollab}}(r, S)$ and $\mathsf{lb}_{\mathrm{pre,nocollab}}(r, S)$, as introduced in equations (23) and (24).

Again, note that within this proposed implementation, the prefix learning phase does not consider the difficulty of the task let to the black-box learning part. For datasets containing inconsistent examples, this could result in sub-optimal overall accuracy in regimes of medium to high transparency, when collaboration between both parts of the Hybrid Interpretable Model is required.

## C.2 HybridCORELS$_{\mathrm{Pre,NoCollab}}$: Empirical Evaluation

We ran the experiments of **Section 5.3** using HybridCORELS$_{\mathrm{Pre,NoCollab}}$ (with the same setup as HybridCORELS$_{\mathrm{Pre}}$), and provide a comparison with HybridCORELS$_{\mathrm{Pre}}$ within Figure 15. The results show that for very low transparency values, HybridCORELS$_{\mathrm{Pre,NoCollab}}$ and HybridCORELS$_{\mathrm{Pre}}$ have very close performances. Indeed, in such regimes, most of the classification task is handled by the black-box part of the model and the absence of collaboration with the interpretable part does not really matter. We observe the same phenomenon in regimes of very high transparency, where most of the examples are classified by the interpretable part. However, in regimes of medium to high transparency, we observe a significant drop of HybridCORELS$_{\mathrm{Pre,NoCollab}}$'s performances. This trend is particularly visible with the ACS Employment dataset. It can be explained by the absence of collaboration between both parts of the model: the prefix learning sacrifices the black-box performances (sending it most of the inconsistent examples) to obtain the most accurate prefix possible. While this policy leads to slightly more accurate interpretable parts compared to the prefixes learned by HybridCORELS$_{\mathrm{Pre}}$, it also harms the overall model accuracy considerably, and the obtained trade-offs are not competitive with those produced by HybridCORELS$_{\mathrm{Pre}}$. As observed in **Section 5.3** with HybridCORELS$_{\mathrm{Pre}}$, on the COMPAS dataset, Hybrid Interpretable Models with intermediate transparency values exhibit better test accuracies than the standalone black-box, due to better generalization. Again, this constitutes an argument in favor of the *Pre-Black-Box* paradigm, as this trend was not observed with the other *Post-Black-Box* methods.

We provide in Figure 16 examples of Hybrid Interpretable Models found with HybridCORELS$_{\mathrm{Pre}}$ and HybridCORELS$_{\mathrm{Pre,NoCollab}}$ on the same data splits of the ACS Employment dataset and transparencies roughly 80%. We observe, as aforementioned, that the black-boxes trained after the HybridCORELS$_{\mathrm{Pre,NoCollab}}$ prefixes exhibit considerably lower performances. On the other hand, the prefix and black-box parts of the models trained using HybridCORELS$_{\mathrm{Pre}}$ have comparable classification

performances, as the former was trained while accounting for the inconsistent samples that would be left for the later to classify.

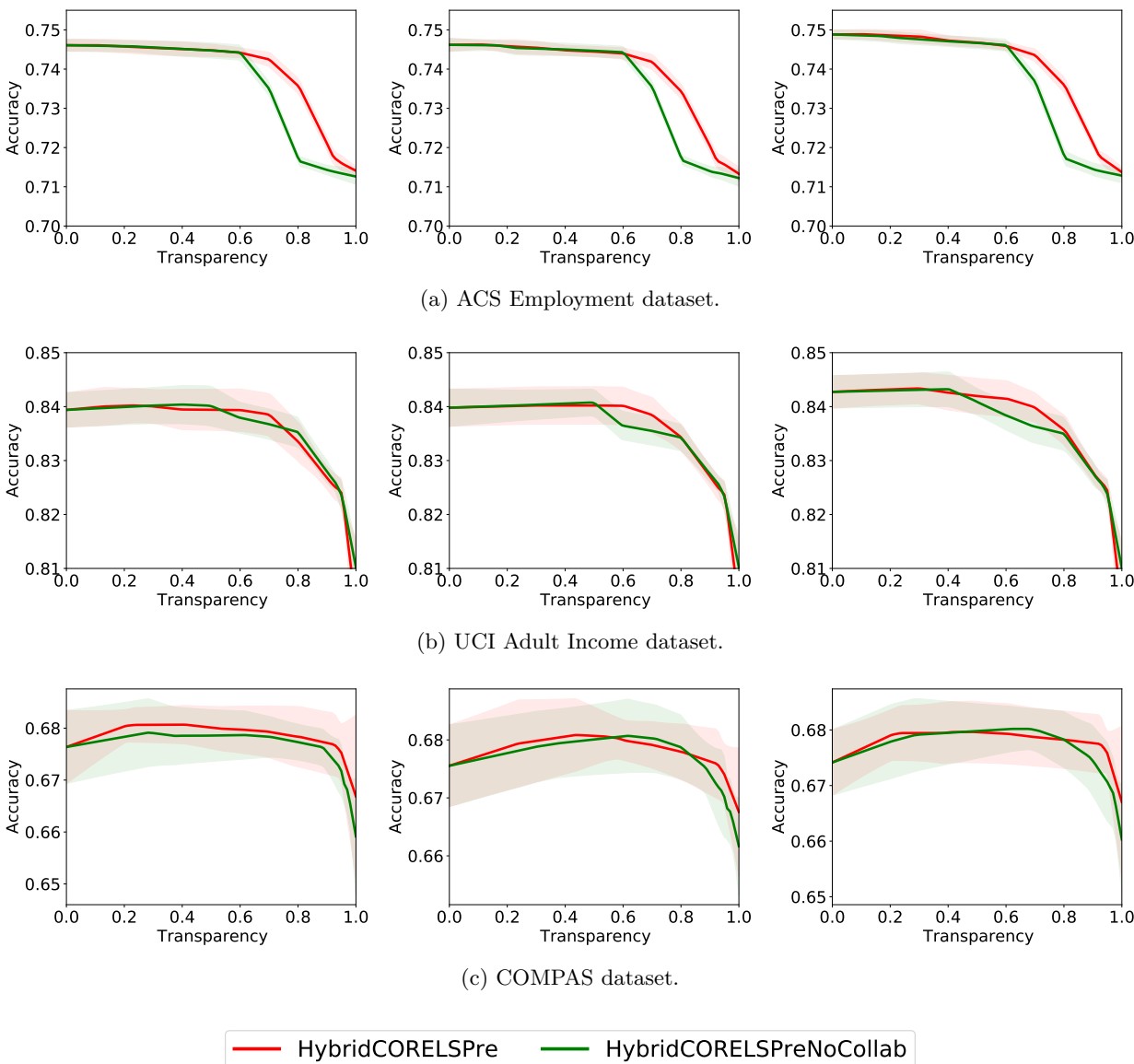

(a) ACS Employment dataset.

(b) UCI Adult Income dataset.

(c) COMPAS dataset.

HybridCORELSPre     HybridCORELSPreNoCollab

Figure 15: Test set accuracy/transparency trade-offs for our two *Pre-Black-Box* variants of HybridCORELS. The Pareto front for each method is represented as a line and the filled bands encode the std across the five data split reruns. Results are provided for several black-boxes: (Left) AdaBoost, (Middle) Random Forests, (Right) Gradient Boosted Trees.

```
if ["age_high" and "Female"] then 0 (acc 71.3%)
else if ["Husband/wife" and "No disability"] then 1 (acc 71.9%)
else if ["age_high" and "Native"] then 0 (acc 64.5%)
else if ["Bachelor's degree" and "No disability"] then 1 (acc 84.8%)
else if ["Reference person" and "No disability"] then 1 (acc 82.3%)
else if ["high school diploma" and "No disability"] then (acc 60.0%)
else
    AdaBoost() (acc 71.9%)
```

(a) HybridCORELS$_{\text{Pre}}$: Test Accuracy 73.7%, Transparency 80.2%.

```
if ["age_low" and "Reference person"] then 1 (acc 82.2%)
else if ["Disability"] then 0 (acc 77.7%)
else if ["age_medium"] then 1 (acc 79.2%)
else if ["age_low" and "Married"] then 1 (acc 69.1%)
else if ["Husband/wife" and "Female"] then 0 (acc 68.6%)
else if ["age_low" and "not own child of householder"] then (acc 59.0%)
else
    AdaBoost() (acc 60.4%)
```

(b) HybridCORELS$_{\text{Pre,NoCollab}}$ : Test Accuracy 71.7%, Transparency 80.3%.

Figure 16: Examples of Hybrid Interpretable Models obtained on the ACS Employment dataset with AdaBoost black-boxes and the same train/validation/test split. The models with transparency closest to 80% were selected. We note that the black-box has worst performance in HybridCORELS$_{\text{Pre,NoCollab}}$ than HybridCORELS$_{\text{Pre}}$ seeing as the prefix sent it the inconsistent examples.

# D  Additional Discussion on the *Pre-Black-Box* Paradigm

Section 5.2 illustrated the impact of the specialization coefficient $\alpha$ used in the *Pre-Black-Box* paradigm. However, these experiments also highlighted a peculiar pattern : the test set performances are often significantly larger than their training set counterparts. For instance, Figure 10 (c) suggests that COMPAS test accuracies can be 5 ppts larger than training accuracies. Similarly, Figure 11 (b) with 0.95 transparency suggests that test set accuracies can be 2 ppts larger than train. How is this possible? Is there data leakage between the training and test sets?

To lay these concerns to rest, we did additional data analysis by comparing the training and test accuracies $1 - \widehat{\mathcal{L}}_S(\langle h_c, r \rangle)/|S|$ of all Hybrid Interpretable Models trained in Section 5.2.

- On COMPAS, 81% of the Hybrid Interpretable Models have a lower test accuracy than training accuracy.

- On Adult, 89% of the Hybrid Interpretable Models have a lower test accuracy than training accuracy.

- On ACS-Employ, 94% of the Hybrid Interpretable Models have a lower test accuracy than training accuracy.

In any cases where the test accuracy was better than training accuracy, the gap was never more than 0.5 ppts, a lot less than the 5 ppts and 2 ppts suggested by Figures 10 & 11. The reason behind these apparent contradictions is that both Figures do not show train/test accuracies of the full tuple $\langle h_c, r \rangle$. Rather, they plot quantities related (but not equivalent) to the full accuracy.

Figure 10 illustrates Upper Bounds on the train/test set accuracies (cf. Equation 16), which sum the errors of the prefix on its support and the unavoidable errors sent to the black-box due to data inconsistencies. Since these are upper bounds (not actual accuracies), the fact that they are higher on test is peculiar, but not indicative of any data leakage. In fact, we compared these Upper Bounds with the actual accuracies on the train/test sets:

- On average, on COMPAS, the upper bounds overestimate train accuracy by 1 ppts and test accuracy by 5 ppts.

- On average, on Adult, the upper bounds overestimate the train accuracy by 2 ppts and the test accuracy by 4 ppts.

- On average, on ACS-Employ, the upper bounds overestimate the train accuracy by 1.6 ppts and the test accuracy by 2.7 ppts.

This explains why, in Figure 10, these upper bounds are better on the test set, while the actual accuracy is better on the training set.

Figure 11 illustrates the train/test accuracy of the black-box evaluated on samples that land in $\overline{\Omega}$ (cf. Equation 17). This quantity is not necessarily representative of the full Hybrid Interpretable Model accuracy for two reasons. First, the full performance must additionally account for the errors made by the prefix over $\Omega$. Second, the black-box training accuracy evaluates the black-box on the training samples that land in $\overline{\Omega}$. However, this region was determined beforehand by minimizing Equation 15 on the **same** training samples. Seeing as Equation 15 penalizes the prefix for making errors on $\Omega$ , it is possible that the hardest training samples (those that could not be predicted by a short rule list) were sent to $\overline{\Omega}$. Consequently, the distributions of training samples landing in $\overline{\Omega}$ is not necessarily the same as the distribution of fresh test samples landing in $\overline{\Omega}$.

