# OpenReview forum: "Learning Hybrid Interpretable Models: Theory, Taxonomy, and Methods"
_TMLR — Accepted by TMLR_

### Review · Reviewer_skY7 · 2024-03-19

**Summary Of Contributions:**

The authors study so-called hybrid interpretable models, that is, the combination of simple rule lists and black-box machine learning models such as random forest. The paper not only provides the theoretical analysis such as the PAC learnability and the error bound but also proposes concrete algorithms and showcases their empirical performance. The proposed methods can precisely control the transparency level (i.e., the ratio of samples classified by a simple model or a black-box model), which was not possible with existing hybrid modeling methods.

**Audience:**

Yes

**Claims And Evidence:**

Yes

**Requested Changes:**

(1) In Section 5.2:
> As expected, higher values of the specialization coefficient $\alpha$ lead to higher training accuracy of the black-box part.

This claim does not seem so obvious in Figure 11. In many cases, at least visually, the difference between small $\alpha$ and large $\alpha$ does not seem significant according to the error bars. I would suggest making the claim more specific.

(2) In Section 5.2, please make the definition of the "improvement rate" clearer.

(3) The comparison in Section 5.3 (Figure 12) is clear for the ACS Employment dataset, but for the other two datasets, the significance of the difference is much less clear. Why is this the case? Some comments will be helpful.

(4) Just out of curiosity, do you have any comments on hybrid modeling where the hypothesis space of the black-box model is infinite? For example, is there a possibility to do some sort of theoretical analysis? This might also be of readers' interest.

**Strengths And Weaknesses:**

Overall I found no major drawbacks. The study is thorough, the topic will surely match the interest of the audience, and the paper is well-written.

\+ The study is thoroughly done from theory to experiment. The theoretical analysis provides meaningful implications of the potential advantage of hybrid models, and the empirical results also support them to some extent.

\+ The taxonomy given for the two different strategies of hybrid modeling is nice. It will be helpful in designing new methods, too.

\- Although the empirical results seem to support the theoretical claims to some extent, the significance of comparisons is sometimes unclear. I list such points in the next form (Requested Changes).

\- The need for the excess length of the paper does not look so obvious. I would suggest trying to streamline the main part even only slightly, e.g. by deferring some empirical results to the appendix.

---

> ### Author Response · Authors · 2024-07-03
> **Response to Reviewer skY7 (1/2)**
>
> Thank you for your valuable and positive feedback! We answer the specific points raised in your review hereafter.
>
> > The need for the excess length of the paper does not look so obvious. I would suggest trying to streamline the main part even only slightly, e.g. by deferring some empirical results to the appendix.
>
> Overall, the revised version of our paper is shorter than the original submission by approximately one page. In particular, we have updated Figure 11 to only display three representative transparency values for each dataset (the results for all datasets, black-boxes and transparency values still being available on the provided supplementary material). Additionally, if deemed necessary by the reviewer, we may keep only the results for a single dataset in the main paper body and defer the results for the two others in the supplementary material. Note that we have also performed a number of other modifications following all the reviewers’ suggestions.
>
> > (1) In Section 5.2: "As expected, higher values of the specialization coefficient lead to higher training accuracy of the black-box part." This claim does not seem so obvious in Figure 11. In many cases, at least visually, the difference between small and large does not seem significant according to the error bars. I would suggest making the claim more specific.
>
> The effect of the specialization coefficient $\alpha$ depends on many factors - such as the transparency level (related to the size of the subset of examples the black-box should actually focus on) or how the examples handled by the black-box are actually more difficult to classify. Because the black-box training algorithms we use do not come up with optimality guarantees, there is no guarantee on the amount of improvement that $\alpha$ will bring on the training or test accuracies on the subset of examples handled by the black-box part. We have added a discussion regarding this aspect at the end of Section 5.2.
> However, we observe that in the wide majority of our experiments (93.33% of all runs) a reasonable (here 2) value of $\alpha$ can improve the test accuracy of the learnt black-box on its assigned subset of examples (and so the test accuracy of the overall hybrid interpretable model).
>
> > (2) In Section 5.2, please make the definition of the "improvement rate" clearer.
>
> We have defined more thoroughly what we mean by “improvement rate” in Section 5.2. More precisely, the improvement rate is the proportion of runs (out of all our performed experiments) for which using a given value of the specialization coefficient $\alpha$ led to an improvement (compared to no specialization at all) of the test accuracy of the learnt black-box (on the subset of examples it classifies) - hence resulting in an improvement of the overall Hybrid Interpretable Model test accuracy, since the prefix part is fixed. An improvement rate of 93.33% (as mentioned in the answer to the previous point) means that in 93.33% of our experiments, setting $\alpha=2$ led to a better accuracy of the resulting hybrid interpretable model at test time, compared to doing no specialization at all.
>
> > (3) The comparison in Section 5.3 (Figure 12) is clear for the ACS Employment dataset, but for the other two datasets, the significance of the difference is much less clear. Why is this the case? Some comments will be helpful.
>
> The results of our main experiments (i.e. the tradeoffs between accuracy and transparency at test time, displayed in Figure 12) show that our method is either as good as the two considered baselines or better. More precisely, it is consistently better on the ACS Employment dataset, which is both the most recent and the largest one. As reported in Section 5.1, the two other datasets (UCI Adult Income and COMPAS) are significantly smaller. This can explain why the generalization error is higher, resulting in a wider standard deviation on the test accuracies over the different considered random seeds. Furthermore, these datasets (in particular COMPAS) contain a substantial number of inconsistent examples - i.e., examples that have the same feature vector but different labels, which hence can never be correctly classified simultaneously. Since the performances of the considered baselines are already close to the best achievable for these two datasets, our method can only be as good as them. These datasets are still widely used in the ML community, but considering other (larger) datasets (such as other ones from the folktables library (https://github.com/socialfoundations/folktables)) would probably result in much more clear differences as is the case with ACS Employment.

---

> > ### Author Response · Authors · 2024-07-03
> > **Response to Reviewer skY7 (2/2)**
> >
> > > (4) Just out of curiosity, do you have any comments on hybrid modeling where the hypothesis space of the black-box model is infinite? For example, is there a possibility to do some sort of theoretical analysis? This might also be of readers' interest.
> >
> > Extending the bound to infinite hypothesis spaces (e.g. linear models) would require leveraging bounds based on the VC-Dimension or the Rademacher Complexity. The end of Section 2.4 now discusses this future work direction.

---

### Review · Reviewer_mhQa · 2024-04-14

**Summary Of Contributions:**

This paper study hybrid models, which is a mixture of interpretable and complex black box models. The authors provide a PAC generalization bound, which gives the indication of a sweet-spot for an optimal level transparency of the hybrid model. Thus, their hybrid models can better balance the level of transparency in the decision making of the model - namely the accuracy-transparency trade-off.

**Audience:**

Yes

**Claims And Evidence:**

Yes

**Requested Changes:**

The paper should be carefully proof-read; language, capitalization and italics are used differently for the same word. Moreover, the format and notation could also be improved as this make the paper tricky to read. I will try to highlight some of the instances below.

In the first sentence of the introduction, the authors could maybe give some references?

Sometimes the authors write "Hybrid Interpretable Models" and sometimes just "Hybrid Models" and "hybrid models".

Explain in more details what these heuristics are? "To address this issue, past studies have optimized such models using local search heuristics (Wang, 2019; Pan et al., 2020)."

Too many parenthesis: "Given the recent development of highly efficient libraries for training interpretable models to optimality (e.g., CORELS for rule-lists (Angelino et al., 2017), GOSDT for decision trees (Hu et al., 2019))),..."

Please put your assumptions into an assumption format. It is not easy to find the assumptions when they are written in text. Do the same for examples and definitions.

Mention in abstract and introduction you consider binary classifiers as it is first stated on page 4.

Some notation is defined several times, and some notation is not defined (only the appendix). E.g., $C_{\Omega}$ is defined several times and $C_{\bar{\Omega}}$ is only defined the appendix.

I think there is a mistake in the definition of $C_{\Omega}$ as is states $x\sim\mathcal{D}$ where $\mathcal{D}$ is the distribution over $\mathcal{X}\times\{0,1\}$.

Make the example if the binary depth-3 decision tree into a example format.

Last sentence on page 7 reads badly.

Something is wrong with "min\_transp"?

**Strengths And Weaknesses:**

The motivation is clear and well stated. I agree with that higher levels of transparency would increasing the trustworthiness of decision-making coming from black box machine learning models. Thus, it is important for a wider adoption of such models.

The paper reads somehow well, but after section 2 the presentation becomes a bit clumsy and harder to follow.

The authors consider rule-based hybrid interpretable models for binary classification; this should be states earlier.

The level of transparency defined as the ratio of samples that are sent to the interpretable part. Can you say something aboutt he ratio of samples that can be explained by the interpretable part?

Another weakness is the lack theoretical content. The PAC bound seems to be quite loose, which seems to give weak indications of this sweet-spot. Therefore, I have troubles linking it practice.

The authors mention that the bound in Thm 1 is loose. Any idea of how to tighten it?

You assume that there exists a perfect model in (1). Please format it as an assumption. How important is this assumption for the analysis of Thm 1?

Instead of ERM, could other principles be used? Like risk-sensitive principles, VaR, CVaR?

The authors shows that a hybrid model trained sufficiently many examples (a finite dataset) can have a PAC-guarantee on new unseen samples. I feel that in this setting it would be more obvious to provide PAC-Bayesian generalizations bounds instead [Alquier, 2024] - where priors could be informative or not.

It is a drawback that the specialization coefficient $\alpha$ has to be fine-tuned in practice. Is there way to come around this? Or do you have some ideas on how to tune it during training?

Pierre Alquier. User-friendly introduction to PAC-Bayes bounds. Foundations and Trends® in Machine Learning,
2024

---

> ### Author Response · Authors · 2024-07-03
> **Response to Reviewer mhQa (1/3)**
>
> Thank you very much for your detailed evaluation of the paper and the insightful comments. We use this response to clarify some of the key points raised in the report.
>
> > The authors consider rule-based hybrid interpretable models for binary classification; this should be states earlier.
>
> We indeed consider rule-based hybrid interpretable models (as defined in Section 3.2) for both our proposed algorithms (Section 4) and the baseline methods (introduced in Section 3.3), and the performed empirical evaluation (Section 5). We have clarified this earlier in the introduction of our paper. However, we emphasize that our theoretical study (Section 2) and taxonomy (Section 3.1) are applicable to any type of hybrid interpretable model (and not only rule-based ones).
>
> > The level of transparency defined as the ratio of samples that are sent to the interpretable part. Can you say something aboutt he ratio of samples that can be explained by the interpretable part?
>
> Transparency is defined (in the literature) as the ratio of samples that are classified by the interpretable part of the hybrid interpretable model. By definition, this part of the model is inherently interpretable, and so all the examples it classifies can be explained. On the other side, the examples classified by the black-box part could be explained using post-hoc explanations on the black-box itself - but the interpretable component of the model is not meant to handle nor explain them (e.g., in rule-based hybrid interpretable models, an example falling into the black-box part was not captured by any rule within the interpretable part, which means the later can not classify nor explain it).
> Note, however, that a thorough explanation for an example classified by the black-box component should first explain why it was not handled by the interpretable part (e.g., because it was not captured by any rule) before explaining the black-box part’s decision.
>
> > Another weakness is the lack theoretical content. The PAC bound seems to be quite loose, which seems to give weak indications of this sweet-spot. Therefore, I have troubles linking it practice.
>
> The aim of our theoretical analysis is not to provide practical bounds, but instead present theoretical evidence for the benefits of sharing inputs between a simple and complex model. To accentuate this point early in the paper, the following text has been added to the Introduction (page 3.)
> “From the theory point of view, we explore Probably-Approximately-Correct (PAC) generalization bound of Hybrid Interpretable Models. The aim of this guarantee is not to provide tight bounds directly relevant to practitioners, but rather to highlight the potential generalization benefits of sharing samples between a complex and simple model. This is evidenced by a sweet spot of the bound’s tightness w.r.t the model transparency“
>
> > The authors mention that the bound in Thm 1 is loose. Any idea of how to tighten it?
>
> We do not think it is possible to tighten it further under the assumptions of the Theorem. Making it tighter would require making more specific assumptions regarding the hypothesis spaces (e.g. using linear models, rule-based models) instead of only assuming that these spaces are finite.
> The bound is loose in general, yes, but it can be tight on very specific examples. The example from Section 2.3 has been updated so that the bound becomes tight for $M=1000$ training samples. Figure 4 has also been updated accordingly.
>
> > You assume that there exists a perfect model in (1). Please format it as an assumption. How important is this assumption for the analysis of Thm 1?
>
> The assumption of perfect model, also known as Realizability Assumption, has been refactored into Assumption 1. This assumption is central to the very definition of PAC-learnability (Definition 3.1, [1]). Relaxing it would lead to Agnostic-PAC-learning guarantees (Definition 3.3, [1]), which is a completely different learning framework that we leave as future work.
>
> > Instead of ERM, could other principles be used? Like risk-sensitive principles, VaR, CVaR?
>
> In principle, we can study any Hölder risk functional applied to the CDF of the loss. Such functionals include the Conditional Value at Risk for instance. Generalization bounds for Hölder risk functionals have been developed recently [2] and are based on the Rademacher Complexity of the hypothesis space and the Lipschitz continuity of the risk functional.
> Applying these bounds to our work would require bounding the Rademacher Complexity (or VC Dimension) of the space $\mathsf{Hyb}$.
>
> [1] Shalev-Shwartz, Shai, and Shai Ben-David. Understanding machine learning: From theory to algorithms. Cambridge university press, 2014.
>
> [2] Leqi, Liu, et al. "Supervised learning with general risk functionals." International Conference on Machine Learning. PMLR, 2022.

---

> ### Author Response · Authors · 2024-07-03
> **Response to Reviewer mhQa (2/3)**
>
> > The authors shows that a hybrid model trained sufficiently many examples (a finite dataset) can have a PAC-guarantee on new unseen samples. I feel that in this setting it would be more obvious to provide PAC-Bayesian generalizations bounds instead [Alquier, 2024] - where priors could be informative or not.
>
> PAC-Bayes might improve our generalization bound in two ways :
> 1) The union bound over all regions $\Omega\in \mathcal{P}$ could be tightened by specifying a prior distribution favoring regions with a certain degree of transparency.
> 2) PAC-Bayes bounds would allow us to bound the risk of infinite hypothesis spaces Hs and Hc (e.g. linear models with few and many coefficients).
>
> Section 2.4 discusses these future work directions.
>
> > It is a drawback that the specialization coefficient $\alpha$ has to be fine-tuned in practice. Is there way to come around this? Or do you have some ideas on how to tune it during training?
>
> The specialization coefficient $\alpha$ is indeed a new hyperparameter which can be used to encourage the black-box part of the model to focus more on its assigned examples during training (since the set of examples assigned to it are predetermined by the interpretable part, learnt first in the Pre-Black-Box paradigm) . It controls a trade-off between two extremes: training only on the subset of examples assigned to it (which would lead to the best training performances but could lead to poor generalization), or training on the entire dataset (which could harm performances in practice, since the black-box may only classify a very small portion of it).
>
> Just like other hyperparameters, fine-tuning the value of this coefficient can be done with the use of a separate validation set, but this process may be computationally expensive. Note however, that (1) only a few set of values should be considered in practice, as our results demonstrate that there is only marginally interest going above $\alpha=3$ or $\alpha=4$ and (2) in the Pre-Black-Box paradigm, retraining the black-box part to fine-tune $\alpha$ does not require retraining the interpretable component. A rule of thumb could be to arbitrarily set $\alpha=2$, since this setting allowed a substantial increase of both the training and test accuracies in almost all of our experiments (93.33% of them, as reported in Section 5.2). Finally, we could also compute a value of $\alpha$ taking into account the transparency level, to make sure the black-box training overall focuses (at least) as much on its assigned subset as on the remaining examples. This is an interesting direction which we now mention as a future work.
>
> > In the first sentence of the introduction, the authors could maybe give some references?
>
> We have included the following examples and references for the use of Machine Learning techniques for high-stakes decision making:
> - kidney exchange (Aziz et al. Optimal kidney exchange with immunosuppressant. AAAI 2021)
> - recidivism prediction (Angwin et al., 2016)
> - credit scoring (Aniceto et al. Machine learning predictivity applied to consumer creditworthiness. Future Business Journal, 2020)
>
> > Sometimes the authors write "Hybrid Interpretable Models" and sometimes just "Hybrid Models" and "hybrid models".
>
> We now use Hybrid Interpretable Models systematically in the manuscript.
>
> > Explain in more details what these heuristics are? "To address this issue, past studies have optimized such models using local search heuristics (Wang, 2019; Pan et al., 2020)."
>
> Additional details have been added to page 2, which now reads
> “ To address this issue, past studies have optimized such models using simulated annealing heuristics (Wang et al., 2019; Pan et al., 2020). More specifically, they employed a rule-set/rule-list as the interpretable component and trained it by randomly adding/removing/permuting rules for a fixed number of steps. Nevertheless, we show empirically that the stochasticity of these simulated annealing heuristic hinders the ability of practitioners to consistently attain a target level of transparency.“
>
> > Too many parenthesis: "Given the recent development of highly efficient libraries for training interpretable models to optimality (e.g., CORELS for rule-lists (Angelino et al., 2017), GOSDT for decision trees (Hu et al., 2019))),..."
>
> We have fixed the parenthesis issue.
>
> > Please put your assumptions into an assumption format. It is not easy to find the assumptions when they are written in text. Do the same for examples and definitions.
>
> As pointed earlier, Assumption 1 has been added in Section 2.2 to clarify the assumption of a mode with perfect accuracy. Additionally, the assumption that one can know $\Omega^*$ in advance has been called Assumption 2. These assumptions are then referenced within the premise of Theorem 2.

---

> > ### Author Response · Authors · 2024-07-03
> > **Response to Reviewer mhQa (3/3)**
> >
> > > Mention in abstract and introduction you consider binary classifiers as it is first stated on page 4.
> >
> > Although our proposed taxonomy is not restricted to the binary classification task, we indeed focus on binary classification for both our theoretical results (Section 2) and proposed methods (Sections 4 and 5), which is in line with the literature, but is now stated clearly in the revised version of our introduction.
> >
> > > Some notation is defined several times, and some notation is not defined (only the appendix). E.g., $C_{\Omega}$ is defined several times and $C_{\overline{\Omega}}$ is only defined the appendix.
> >
> > The official definitions of transparency ($C_{\Omega}$) and opacity ($C_{\bar{\Omega}}$) are now presented as Definition 1 in Section 2.1.
> >
> > > I think there is a mistake in the definition of $C_{\Omega}$ as is states $x \sim \mathcal{D}$ where is the distribution over $\mathcal{X} \times \{0,1\}$.
> >
> > Thank you for pointing this out! The definition has been corrected and put in Definition 1, see the point above.
> >
> > > Make the example if the binary depth-3 decision tree into a example format.
> >
> > The text now reads like an example.
> >
> > > Last sentence on page 7 reads badly.
> >
> > We have fixed this by rephrasing this sentence.
> >
> > > Something is wrong with "min_transp"?
> >
> > We have updated this notation, replacing min_transp by a more thorough $C_{\text{min}}$.

---

### Review · Reviewer_ufqm · 2024-06-18

**Summary Of Contributions:**

This paper introduces a new framework to think about hybrid interpretable models, which send a fraction of the data that lies in a fixed domain to a simpler, more transparent model, and the remainder to a more complex black-box model. The paper provides a theoretical analysis, identifying two families of learning methods (pre- and post-), and a new algorithm for learning rule-based hybrid models. The paper evaluates the new algorithm (both pre- and post- versions) on three public datasets, showing improved performance over certain existing algorithms.

**Audience:**

Yes

**Broader Impact Concerns:**

A broader impact statement has not been provided. My main concerns with impact are summarized in point 2, regarding practical usage of hybrid models, and providing explanations for decisions that are routed through the black-box.

I would also recommend that the authors think about other ethical concerns and add a broader impact section (e.g., cautioning against premature use of their work; poor choices of pre-mined rules which could cause discrimination, etc.), since their work lies in the domain of Fairness, Accountability, Transparency and Explainability of AI models.

**Claims And Evidence:**

Yes

**Requested Changes:**

What follows is a list of moderate and minor changes. I would look to the authors' responses to comments 1-3, 10, 14-16, and 23-26, while making my recommendation.

1-4. As above.

5. (p.1, third line from bottom) "worst" should be "worse"
6. (p.3) Clarify that HybridCORELS provides a certificate of optimality in terms of accuracy
7. (Fig 2b) Missing mathcal in caption for H_s
8. (Sec 2.3) It is unclear to me why an average over epsilon provides a measure of "looseness" of the bound. The optimal transparency isn't obtained at the point where the bound is least loose, it is obtained at the point where the bound is smallest (which is what is effectively done in Fig 4a). The word "looseness" feels misleading to me. Is there a better nomenclature?
9. (Sec 2.4, point 3) Whenever the authors say that hybrid models may obtain better generalization, this sounds very speculative. They should at least mention the fact that they see this in their experiments in Section 5.3.
10. (Sec 2.4) Also, can the authors provide some intuition as to why the optimal transparency is non-zero?
11. (Sec 3.1, 2nd paragraph) Second sentence grammar
12. (Sec 3.1.1) The claim that the interpretable model is able to "correct the mistakes" made by the black-box sounds non-intuitive and should be qualified. One normally expects the black-box model to be more accurate.
13. (Sec 3.1.1 and 3.1.2) The overall writing here feels very speculative and discussion-heavy - for example, "however, it could also cause overfitting". the discussion is not balanced since there is no mention of the lack of expressivity of the transparent model. Reading this, one is made to feel that the black box is less accurate and also less transparent. It would be better to stick to describing the approach, and maybe have a separate section to provide intuitions/discussion.
14. (Sec 3.3.1) Criticisms of HyRS should be toned down. It is unclear why the objective is considered "extremely complex". One strategy to obtain a given transparency could be to run the algorithm for different beta until we obtain the desired transparency - why is this not a possibility? Also, isn't some degree of variance normal in ML, and can't we choose the run with greatest accuracy?
15. (Sec 3.3.2) The paper states that for CRL, one obtains transparencies of 0.75 when keeping a target of 0.5, and that this may affect accuracy. Is a decrease in accuracy observed in practice? What are the accuracies of the points in Fig 9?
16. For both HyRS and CRL, there are significant "jumps" in the transparency on the UCI Adult dataset. The part where transparency jumps in Fig 7b is probably because some rule becomes "active" at that beta value, and captures around half the dataset. Does HybridCORELS achieve fewer jumps in Fig 12? If not, then these criticisms should be avoided.
17. (p.13, line 2 and line 4) "counts" should be "count"; "features vector" should be "feature vectors"
18. (Sec 5.1) Why are a pre-mined set of rules used, instead of using a strategy similar to a decision tree?
19. (Sec 5.2) Please separate _observations_ about the results from _interpretations_ drawn.
20. (Sec 5.2, results section, 2nd paragraph) There are some conflicting statements here: first it is claimed that higher alpha is better (which is indeed seen from the figures). Then, it is justified that specializing the black-box on the samples not sent to the transparent model is a good thing. This undercuts the need to introduce alpha in Section 3.1.2 in the first place, because now overfitting is considered to be good. Then, later in the paragraph, it is claimed that a small alpha (around 1) is good, because it maximizes the "improvement rate". But it is not clear to me why we care about improvement rate - shouldn't we care more about the raw hybrid test accuracy?
21. (Fig 10) Is the variance in the plots over multiple runs or over the 3 different black box models? This has not been specified. If over multiple runs, then which black-box model is shown, and can we see the others? If the variance is over models, then can we see the variance over multiple runs?
22. (Fig 10) It would be useful to see the performance at 100% transparency as well, as a baseline.
23. (Fig 10) Why (and how) is test accuracy consistently higher than training accuracy?
24. (Fig 11) Once again, test accuracy is higher than training accuracy in many instances - how is this possible?
25. (Fig 11) Taking an optimal value of alpha from these plots does not appear well-motivated, since there is a lot of noise and no clear trend. The main trend appears to be higher accuracy for higher alpha. In any case, alpha should be optimized through cross-validation in the hybrid model (as mentioned in point 1)
26. (Fig 12) For different fixed transparency values, it would be very useful to understand where all of these models fall on the precision-recall curve, not just understanding accuracy (i.e., recall).
27. (Sec 6, line 1) Please cite.

**Strengths And Weaknesses:**

One extremely interesting result obtained by this paper is that hybrid models can have better accuracies than more complex black-box models that have no transparency. Hybrid models are thus a pareto improvement on black-box models in terms of both accuracy and transparency. This is a surprising result, and the authors provide some theoretical justification for it, and also show this empirically in (at least) one of the three datasets they examine.

This finding is a great argument for further study of hybrid models, which the paper claims is an under-explored area.

In my view, the paper does not have very serious weaknesses - a summary of moderate to minor issues follows, with more detail in the next section. From a technical standpoint:

1. The lack of cross-validation over the specialization coefficient, alpha, is a statistical issue. Currently, the authors choose the value of alpha based on both training and test data (presumably the entire dataset), and then use this value while evaluating against other methods - this bleeds test data into the model, creating bias at the evaluation stage. Alpha should be treated like any other hyperparameter and be optimized through cross-validation.

2. An important question I have about hybrid models in general: what if we need an explanation for a sample that is sent to the black-box? For example, if the model is designed to accept or reject applications for a loan, how would we explain rejection decisions that are routed through the black-box? This is particularly problematic if the transparent model would have disagreed with the black-box model on this sample. If we choose to default to the decision of the transparent model in all cases of disagreement, then this would make the hybrid model strictly worse than the transparent model. How do the authors imagine hybrid models to be used in practice?

From a stylistic standpoint:

3. There are several claims of being "thorough", which should be dialed back. The paper undertakes an exploration of an understudied area, so there are likely a very large number of possible approaches. The authors have made choices to restrict their analysis to a specific subset of approaches, which is fine. But claims that this foray is comprehensive are misleading and should be omitted.

4. The writing quality is good, but feels quite verbose and repetitive. Often, discussion-style comments are interwoven throughout other sections, including sentences of a more speculative nature. This makes it difficult to extract the signal from the noise. I would encourage the authors to make efforts to condense the writing where possible, and to separate raw observations from interpretations and speculation.

---

> ### Author Response · Authors · 2024-07-03
> **Response to Reviewer ufqm (1/6)**
>
> Thank you for all your detailed comments and recommendations on this paper. We address all the points raised in your review hereafter.
>
> > 1. The lack of cross-validation over the specialization coefficient, alpha, is a statistical issue. Currently, the authors choose the value of alpha based on both training and test data (presumably the entire dataset), and then use this value while evaluating against other methods - this bleeds test data into the model, creating bias at the evaluation stage. Alpha should be treated like any other hyperparameter and be optimized through cross-validation.
>
> In Section 5.2, we perform experiments using only HybridCORELSPre, to empirically assess the effectiveness of the specialization coefficient $\alpha$ to improve the black-box accuracy on the subset of examples it effectively handles (at train and test time). For these experiments, we split each dataset into training (80%) and test (20%) sets. Overall, we conclude that a value of $\alpha=1$ allows for a substantial improvement most of the time, and fix this value for the remainder of the paper.
>
> In Section 5.3, we then compare our methods with state-of-the-art rule-based hybrid interpretable models learning algorithms. For these experiments, we keep $\alpha=1$ for HybridCORELSPre. However, each dataset is split into training (60%), validation (20%) and test (20%) sets - which means that the training and test sets are different from that used in Section 5.2, and so are the learnt models (and the test data of these new models is never used to determine any hyperparameter).
>
> Optimizing the value of $\alpha$ using the separate validation set (as is done for the other hyperparameters) would be possible, and would likely improve the results, since sticking to one configuration does not allow adapting to the different scenarios. We chose to not do so and let the choice of $\alpha$ as an interesting research avenue, in order to keep the size of the hyperparameters’ grid comparable among the different methods. We clarify these aspects at the end of the “Pre-Black-Box method setup” paragraph of Section 5.3.
>
> > 2. An important question I have about hybrid models in general: what if we need an explanation for a sample that is sent to the black-box? For example, if the model is designed to accept or reject applications for a loan, how would we explain rejection decisions that are routed through the black-box? This is particularly problematic if the transparent model would have disagreed with the black-box model on this sample. If we choose to default to the decision of the transparent model in all cases of disagreement, then this would make the hybrid model strictly worse than the transparent model. How do the authors imagine hybrid models to be used in practice?
>
> Indeed, in high-stakes settings where one would default to the interpretable decision in case of disagreements, the black-box is not useful and so should be completely discarded. In alternative settings where imperfect explanations are sometimes acceptable, one could explain black-box predictions with a post-hoc method such as SHAP. Importantly, recent work has demonstrated that SHAP attributions can be more faithful to the model if it is restricted to regions of the input space [1]. Thus, it might be possible to more faithfully explain the black-box in its domain of application $\bar{\Omega}$. This is left as future work.
>
> > 3. There are several claims of being "thorough", which should be dialed back. The paper undertakes an exploration of an understudied area, so there are likely a very large number of possible approaches. The authors have made choices to restrict their analysis to a specific subset of approaches, which is fine. But claims that this foray is comprehensive are misleading and should be omitted.
>
> This point was also raised by Reviewer mhQa. On the one hand, the theoretical study (Section 2) and taxonomy (Section 3.1) parts are applicable to any type of hybrid interpretable model (and not only rule-based ones). On the other hand,  we indeed specifically consider rule-based hybrid interpretable models (as defined in Section 3.2) for both our proposed algorithms (Section 4) and the baseline methods (introduced in Section 3.3), and the performed empirical evaluation (Section 5). We have clarified this earlier in the abstract and introduction of our paper.
>
> [1] Laberge, Gabriel, et al. "Tackling the XAI Disagreement Problem with Regional Explanations." International Conference on Artificial Intelligence and Statistics. PMLR, 2024.

---

> ### Author Response · Authors · 2024-07-03
> **Response to Reviewer ufqm (2/6)**
>
> > 4. The writing quality is good, but feels quite verbose and repetitive. Often, discussion-style comments are interwoven throughout other sections, including sentences of a more speculative nature. This makes it difficult to extract the signal from the noise. I would encourage the authors to make efforts to condense the writing where possible, and to separate raw observations from interpretations and speculation.
>
> Section 3.1.3 that speculates about the “end-to-end” paradigm has been moved to the discussion. We think this section fits better in future work discussions than in the core of the paper. Moreover, many repetitive sentences have been removed/shortened. Overall, the revised manuscript is shorter than the submitted version by an entire page.
>
> > 5. (p.1, third line from bottom) "worst" should be "worse"
>
> Thank you for pointing this out, we fixed this typo.
>
> > 6. (p.3) Clarify that HybridCORELS provides a certificate of optimality in terms of accuracy
>
> We have updated the listing of our contribution at the end of the introduction to make clear that HybridCORELS provides a certificate of optimality in terms of accuracy, for a given transparency constraint.
>
> > 7. (Fig 2b) Missing mathcal in caption for H_s
>
> Thank you for pointing this out, we fixed this.
>
> > 8. (Sec 2.3) It is unclear to me why an average over epsilon provides a measure of "looseness" of the bound. The optimal transparency isn't obtained at the point where the bound is least loose, it is obtained at the point where the bound is smallest (which is what is effectively done in Fig 4a). The word "looseness" feels misleading to me. Is there a better nomenclature?
>
> Indeed, the word “looseness” is not adequate since it implies that the “least loose” bound is itself “loose” or “uninformative”. That is not the case, as evidenced by our new Figure 4c) showing that the optimal bounds are not loose on the toy example.
> We have changed the nomenclature to “Average Bound”, which describes exactly what $\overline{\mathcal{B}}$ is.
>
> > 9. (Sec 2.4, point 3) Whenever the authors say that hybrid models may obtain better generalization, this sounds very speculative. They should at least mention the fact that they see this in their experiments in Section 5.3.
>
> Sec 2.4 now refers to Figure 12 (c) when claiming that hybrid models can improve generalization.
>
> > 10. (Sec 2.4) Also, can the authors provide some intuition as to why the optimal transparency is non-zero?
>
> Under the assumption that a perfect model exists within the class (Assumption 1 in the updated manuscript), errors made by any learned model must be caused by “overfitting”. The non-zero optimal transparencies therefore imply that sending black-box samples to the simple model reduces overfitting (aka sharing samples is a form of regularization). These conclusions hold within the assumptions of the Theorem no they might not always hold in practice.
>
> > 11. (Sec 3.1, 2nd paragraph) Second sentence grammar
>
> Thank you for pointing this out, we fixed the formulation.
>
> > 12. (Sec 3.1.1) The claim that the interpretable model is able to "correct the mistakes" made by the black-box sounds non-intuitive and should be qualified. One normally expects the black-box model to be more accurate.
>
> We indeed expect the black-box model to be more accurate. What we meant here is that in the Post-Black-Box paradigm, the black-box is learnt first, and then its parameters are fixed. Because the interpretable model is fitted afterwards, it can be trained with the knowledge of the black-box mistakes, and specifically try to capture the corresponding examples and classify them correctly. In such a case, it effectively corrects these misclassifications from the perspective of the overall hybrid interpretable models. This is an important advantage of Post-Black-Box hybrid interpretable models learning algorithms, pointed out in the related work (including in the papers proposing HyRS and CRL). We have reformulated our writing to clarify this aspect.
>
> > 13. (Sec 3.1.1 and 3.1.2) The overall writing here feels very speculative and discussion-heavy - for example, "however, it could also cause overfitting". the discussion is not balanced since there is no mention of the lack of expressivity of the transparent model. Reading this, one is made to feel that the black box is less accurate and also less transparent. It would be better to stick to describing the approach, and maybe have a separate section to provide intuitions/discussion.
>
> We have rewritten Sections 3.1.1 and 3.1.2 to stick to the descriptions of the approaches.

---

> ### Author Response · Authors · 2024-07-03
> **Response to Reviewer ufqm (3/6)**
>
> > 14. (Sec 3.3.1) Criticisms of HyRS should be toned down. It is unclear why the objective is considered "extremely complex". One strategy to obtain a given transparency could be to run the algorithm for different beta until we obtain the desired transparency - why is this not a possibility? Also, isn't some degree of variance normal in ML, and can't we choose the run with greatest accuracy?
>
> We removed the claim in Section 3.3.1 that the objective of HyRS is “extremelly complex” and instead accentuate that using a soft constraint on transparency requires that the user perform a line-search over $\beta$ to get the desired transparency. We have added Figures 9 c) and d) to highlight the benefits of using a hard constraint instead.
>
> > 15. (Sec 3.3.2) The paper states that for CRL, one obtains transparencies of 0.75 when keeping a target of 0.5, and that this may affect accuracy. Is a decrease in accuracy observed in practice? What are the accuracies of the points in Fig 9?
>
> We have simply removed any claim regarding accuracy from Sec 3.3.2. The goal of these experiments is simply to show that it can be hard to get a specific transparency. All discussions on trade-offs between accuracy and transparency are left to Section 5.3.
>
> > 16. For both HyRS and CRL, there are significant "jumps" in the transparency on the UCI Adult dataset. The part where transparency jumps in Fig 7b is probably because some rule becomes "active" at that beta value, and captures around half the dataset. Does HybridCORELS achieve fewer jumps in Fig 12? If not, then these criticisms should be avoided.
>
> We have added Figure 9 c) and d) to show that HybridCORELS does not exhibit the large jumps that were seen in HyRS/CRL.
>
> > 17. (p.13, line 2 and line 4) "counts" should be "count"; "features vector" should be "feature vectors"
>
> Thank you for pointing this out, we fixed these typos.
>
> > 18. (Sec 5.1) Why are a pre-mined set of rules used, instead of using a strategy similar to a decision tree?
>
> State of the art (scalable) methods for learning optimal rule lists all require rules pre-mining, because jointly solving the rule generation and rule ordering problem results in a large, difficult optimization problem which is not efficiently handled by existing approaches. Greedy algorithms (such as CART for decision trees) can be used to generate rule lists, but we chose not to do so, for several reasons. First, they would only allow the generation of suboptimal results, and the solutions would come without any optimality guarantee. Second, greedy learning of rule lists models is tricky: contrary to decision trees, once a rule is added to the model, it directly classifies a subset of the training set, and there is no way to “undo” this decision or “refine” the rule (contrary to a decision tree, in which sub-branches can still be added to refine partitioning). This leads to rule lists models with poor performances, as was already observed in this recent work (in the context of privacy-protected rule lists): Ly et al. Smooth Sensitivity for Learning Differentially-Private yet Accurate Rule Lists. 2024
>
> Finally, these two points are even strengthened by the fact that we enforce an additional constraint (on the prefix coverage), and in such a case performing global (rather than greedy) optimization is expected to yield better results. We have added a few words regarding the choice of CORELS (which relies on a set of pre-mined rules) at the end of Section 4.1.
> The fact that we use sets of pre-mined rules is additionally required by the two state-of-the-art methods we consider in our experiments and eases a fair comparison with our proposed approaches.
>
> > 19. (Sec 5.2) Please separate observations about the results from interpretations drawn.
>
> We have modified the “Results” paragraph to make sure there is no confusion between the observations made from the reported plots, and the interpretations we propose to them.
>
> > 20. (Sec 5.2, results section, 2nd paragraph) There are some conflicting statements here: first it is claimed that higher alpha is better (which is indeed seen from the figures). Then, it is justified that specializing the black-box on the samples not sent to the transparent model is a good thing. This undercuts the need to introduce alpha in Section 3.1.2 in the first place, because now overfitting is considered to be good. Then, later in the paragraph, it is claimed that a small alpha (around 1) is good, because it maximizes the "improvement rate". But it is not clear to me why we care about improvement rate - shouldn't we care more about the raw hybrid test accuracy?
>
> Higher values of $\alpha$ should always lead to higher training accuracy of the black-box component on the subset of training examples that are sent to it, as increasing alpha directly increases the cost of misclassifying these examples in the objective function. [...]

---

> ### Author Response · Authors · 2024-07-03
> **Response to Reviewer ufqm (4/6)**
>
> [...] Since the interpretable part is trained first in the Pre-Black-Box paradigm, its predictions do not depend on $\alpha$, and the training accuracy of the whole hybrid interpretable model is also improved when increasing $\alpha$.  The key point is then: does increasing the training accuracy also increase the test accuracy - and after what value of $\alpha$ does overfitting occur (if any ?). We thus report the improvement rate, which is the proportion of experiments on which a given value of alpha led to a higher test accuracy (of the black-box model, on its assigned examples and thus of the whole hybrid interpretable model as discussed previously) than no specialization at all (i.e., $\alpha=0$). In particular, and as stated in the text (Section 5.2, Results paragraph), when $\alpha=2$, this improvement rate is 93.33%. This means that among all the performed experiments, setting $\alpha=2$ (rather than not specializing the black-box at all) led to an increase of the whole hybrid interpretable model accuracy in 93.33% of them, which confirms the usefulness of this parameter when set to a “reasonable” value.
>
> Note that values of $\alpha$ greater of 2 lead to smaller improvement rates (for instance, for the same set of experiments, the improvement rate is 73.33% for $\alpha=5$, and only 66.6% for $\alpha=8$ - which suggests that overfitting progressively occurs for too large values of $\alpha$). We included these additional numbers within Section 5.2, and we have clarified the definition of “improvement rate” in Section 5.2 as suggested by Reviewer skY7.
>
> > 21. (Fig 10) Is the variance in the plots over multiple runs or over the 3 different black box models? This has not been specified. If over multiple runs, then which black-box model is shown, and can we see the others? If the variance is over models, then can we see the variance over multiple runs?
>
> The variance in the plots of Figure 10 is measured over five runs with different random seeds. For each of these five experiments, we follow the same experimental method: we learn a prefix using different hyperparameters within HybridCORELSPre, and retain the prefix yielding the highest training accuracy upper bound (such upper bounds being computed as the sum of the errors made by the prefix over the examples it classifies and of the minimum number of errors a black-box can not avoid due to inconsistencies in the data not handled by the prefix). We have clarified this point in the caption of Figure 10. Importantly, note that at this stage, we have only built prefixes (and not any associated black-box - hence the computation of an accuracy upper bound).
>
> > 22. (Fig 10) It would be useful to see the performance at 100% transparency as well, as a baseline.
>
> We do not compute results at 100% transparency within Section 5.2 because the objective of this subsection is to characterize the effect of the specialization coefficient $\alpha$, which aims at specializing the black-box on the subset of examples that are assigned to it (and if all examples are handled by the interpretable component the notion of black-box specialization is no longer useful since the black-box remains un-used). We rather focus on a limited number of transparency constraints values (namely low transparency (0.25), medium transparency (0.5), high transparency (0.75, 0.85) and very high transparency (0.95), as stated in the “Setup” paragraph). Note that results at 100% transparency are provided in the following Section (5.3) in which we study the accuracy performances of the whole hybrid interpretable models.
>
> > 23. (Fig 10) Why (and how) is test accuracy consistently higher than training accuracy?
>
> Note that we report in Figure 10 the (train or test) accuracy upper bound of an hybrid interpretable model built from the learnt prefix (and not an absolute accuracy). It can be decomposed into two parts: the number of misclassification errors made by the learnt prefix (on the subset of examples it captures), and the number of misclassifications the black-box cannot avoid (on the subset of examples left to it) due to inconsistencies in the training data.
>
> Decomposing the accuracy upper bounds displayed in Figure 10 is interesting. For instance, consider the COMPAS dataset experiments (Figure 10(c)). At 0.25 transparency, the training accuracy upper bound is approximately 0.71 while the test accuracy upper bound is above 0.75 (i.e., significantly better). This means that any Hybrid Interpretable Model built from the associated prefixes can have at most a training accuracy of 0.71 (overall) and a test accuracy of 0.75 (overall). [...]

---

> ### Author Response · Authors · 2024-07-03
> **Response to Reviewer ufqm (5/6)**
>
> [...] However, the training accuracy of the prefix is about 0.73 while its test accuracy is slightly smaller, around 0.73 as well. The difference in the accuracy upper bounds then results from a difference in the second term: the proportions of errors (due to data inconsistencies) that the black-box cannot avoid. Indeed, the black-box accuracy upper bound is above 0.76 on the test set, while it is below 0.71 on the training set. This difference can be explained by two factors: either the train set contains more inconsistent examples (which is possible since it is larger), or more inconsistent examples are left (in proportion) to the black-box component in training than in test. In both cases, this trend is complex and further analysis can provide interesting insights.
>
> > 24. (Fig 11) Once again, test accuracy is higher than training accuracy in many instances - how is this possible?
>
> Figure 11 reports train and test accuracies of the black-box part of the built hybrid interpretable models (the prefix/interpretable part being constant as it is learnt before the black-box). The train accuracy is better than the test accuracy in most of the plots (for instance, for all values of the specialization coefficient and all transparency values for the experiments using the ACS Employment dataset (Figure 11 (a)). For some experiments (mainly two: the low transparency plots for the UCI Adult Income and COMPAS datasets), the test accuracy is better than the train accuracy. Since this low transparency regime corresponds to the scenario where the black-box is trained to classify most of the dataset’s examples, it makes sense that the black-box generalizes the best in this situation. The difference between train and test accuracy for these experiments is rather small, and typically lies in the standard deviation interval (and these are the two smallest datasets used in our experiments).
>
> > 25. (Fig 11) Taking an optimal value of alpha from these plots does not appear well-motivated, since there is a lot of noise and no clear trend. The main trend appears to be higher accuracy for higher alpha. In any case, alpha should be optimized through cross-validation in the hybrid model (as mentioned in point 1)
>
> As discussed in our answer to Point 1., we totally agree that $\alpha$ could be optimized using cross-validation, and this would likely result in better results for HybridCORELSPre. While there is no clear trend on the effect of $\alpha$ on the black-box performances which would hold for all considered scenarios (transparency values and datasets), we still observe that the improvement rate (proportion of experiments for which a particular value of $\alpha$ led to best test performances) was the highest for values of 1 or 2, as reported in the main text (paragraph “Results” of Section 5.2). Higher values of $\alpha$ lead to smaller improvement rates, although, of course, optimizing $\alpha$ separately for each experiment (on a separate validation set) would result in a finer choice of the optimal value based on the instance’s characteristics. This relates to Point (2) of Reviewer skY7: we have clarified the definition of “improvement rate” in Section 5.2 and how it was used to fix the value of $\alpha$ for further experiments.
>
> > 26. (Fig 12) For different fixed transparency values, it would be very useful to understand where all of these models fall on the precision-recall curve, not just understanding accuracy (i.e., recall).
>
> Future work should definitely investigate the precision-recall of Hybrid Interpretable Models. Yet for this work, since our datasets are well-balanced (COMPAS classes ratio is 0.52:0.48, Adult 0.76:0.24, ACS-Employ 0.42:0.58) we argue that accuracy is a good metric to compare various models.
>
> > 27. (Sec 6, line 1) Please cite.
>
> References to prior work on Hybrid Interpretable Models have been added to the conclusion.

---

> > ### Author Response · Authors · 2024-07-03
> > **Response to Reviewer ufqm (6/6)**
> >
> > > A broader impact statement has not been provided. My main concerns with impact are summarized in point 2, regarding practical usage of hybrid models, and providing explanations for decisions that are routed through the black-box.
> > I would also recommend that the authors think about other ethical concerns and add a broader impact section (e.g., cautioning against premature use of their work; poor choices of pre-mined rules which could cause discrimination, etc.), since their work lies in the domain of Fairness, Accountability, Transparency and Explainability of AI models.
> >
> > We have added a Broader Impact Statetement that reads:
> >
> > “Hybrid Interpretable Models explore the accuracy-transparency trade-offs, but they do not address the lack of explainability of black-box models. As a result, Hybrid Interpretable Models are not adequate for critical systems where explanations are required for any model decision. This is because the few decisions relayed to the black-box cannot be faithfully explained. Rather, Hybrid Interpretable Models are potentially useful for tasks where imperfect explanations are sometimes acceptable. In such settings, the interpretable component can be used to get the "big picture'' of the model behavior, while post-hoc methods can be used to get imperfect explanations of the black-box on "edge cases''.
> >
> > Moreover, a rule-based model is only as good as its premined rules. A poor collection of rules can cause performance degradation but also discrimination. Indeed, as evidenced by Figure 13, rule-based models can discriminate based on protected attributes such as gender and ethnicity. Consequently, we caution against the premature use of our work and advocate that practitioners exclude protected attributed from their rules before applying HybridCORELS.”

---

> > > ### Comment · Reviewer_ufqm · 2024-07-19
> > > **Response to authors**
> > >
> > > I thank the authors for taking the time to carefully respond to each of my comments.
> > >
> > > I will continue to use the same numbering scheme as in my review to respond to the authors' responses. I respond to points 23 and 24 first because I now think these are more serious issues than points 1 and 20.
> > >
> > >
> > > 23. Based on my understanding of the authors' response, they are essentially saying that test accuracy is greater than training accuracy purely by chance, because the test set happens to have a larger fraction of "easier" examples than the training set (I am paraphrasing - please correct me if I misunderstood). I am suspicious of this argument because we see it in all three datasets in Fig 10. The trend of higher test accuracy is highly consistent, which is worrisome, and makes the reader feel that there could be bugs in the code. Could the authors please evaluate the test accuracy using 5-fold cross validation? This should ensure that all parts of the data are being used evenly, preventing only easy examples from landing in the test set. Failing that, the authors would need to provide a much more robust explanation for why test accuracy is consistently higher.
> > >
> > >
> > > 24. I am somewhat unconvinced by the authors' argument that the black-box will *generalize* better because it is trained on a larger number of examples. While one may expect the black-box to *perform* better (which it does, based on the accuracies for different levels of transparency), I do not believe it should be able to achieve higher accuracy on the test set. My understanding is that these datasets are large enough that there would be some overfitting on the training data. Furthermore, if the errorbars show standard deviation (and not standard error of the mean), then the difference could be statistically significant (once you divide by square-root of N), and is consistently present across all values of $alpha$ for the two plots mentioned.
> > >
> > >
> > > I believe these two points represent very basic sanity-checks on the analysis, which the paper is currently not passing.
> > >
> > > ---
> > >
> > > 1. I feel that the authors have not understood my issue in this point. Let's say that the dataset is divided into three parts $D_{train}$, $D_{val}$ and $D_{test}$ (at a 60-20-20 ratio) for training, validation and testing respectively. Then, in Section 5.2, for choosing $\alpha$, the authors have used an 80-20 split (of the full dataset, I assume) - so let us say they use $D_{train}$ and $D_{val}$ for training and $D_{test}$ for testing. Finally $\alpha$ is chosen by looking at the accuracies on _all_ these parts of the dataset. In other words, the estimate $\hat\alpha$ that the authors arrive at depends on $D_{train}$, $D_{val}$ and $D_{test}$.
> > >
> > >     Now, in Section 5.3, when they fit the model on just $D_{train}$, they are using $\hat\alpha(D_{train}, D_{val}, D_{test})$ during fitting, so in other words, *they are indirectly using information from $D_{test}$ to train the model*. The authors can address this in one of three ways: (1) reproduce the figures in Section 5.2 using only $D_{train}$ and $D_{val}$ (i.e., not using $D_{test}$ while choosing $\alpha$); (2) cross-validate to determine $\alpha$ along with other hyper-parameters; or (3) acknowledge this issue in the paper itself state that they do not expect the results to be significantly different because their method for choosing $\alpha$ is fairly heuristic.
> > >
> > > ---
> > >
> > > 20. Firstly, in accordance with point 1 above, let us distinguish between test and validation accuracy, because $\alpha$ is a hyperparameter and should be chosen based on training and validation data (not test data). Now, I understand that the "improvement rate" is the percentage of experiments in which you get higher validation accuracy, but I am trying to understand why this is a better metric than the average validation accuracy across experiments. Is it because there is very high variance? Can the authors provide a plot of the improvement rate as a function of $\alpha$?
> > >
> > >     Furthermore, if the improvement rate is highest for $\alpha = 2$, why have the authors chosen to use $\alpha = 1$ in Section 5.3? This feels very arbitrary, and reduces reader confidence in the whole analysis.
> > >
> > >     Finally, is there any ML literature that could recommend or support using a quantity like improvement rate to choose hyper-parameters in hybrid interpretable models, or is this an innovation presented in this paper?
> > >
> > >
> > > I acknowledge and thank the authors for their replies to all of my other comments, as well as the edits made to the manuscript.

---

> > > > ### Comment · Action_Editor_xF5u · 2024-07-19
> > > >
> > > > Thanks to the reviewers and authors for discussing the paper.
> > > >
> > > > All reviewers have submitted their recommendation, but one reviewer suggests they are not convinced that the testing error is more favourable than the training error. I think this is quite an interesting thing to happen, especially across multiple settings. Before we make our final decision on the paper, it would be great if the authors could respond to any remaining feedback, with a special focus on the training/testing evaluation.

---

> > > > ### Author Response · Authors · 2024-07-29
> > > > **Response to Reviewer ufqm (1/2)**
> > > >
> > > > Thank you for your follow-up and for your detailed evaluation of our paper. We address hereafter your different concerns and hope this will help clarify the details of our experimental setup.
> > > >
> > > >
> > > > > 23. Based on my understanding of the authors' response, they are essentially saying that test accuracy is greater than training accuracy purely by chance, because the test set happens to have a larger fraction of "easier" examples than the training set (I am paraphrasing - please correct me if I misunderstood). I am suspicious of this argument because we see it in all three datasets in Fig 10. The trend of higher test accuracy is highly consistent, which is worrisome, and makes the reader feel that there could be bugs in the code. Could the authors please evaluate the test accuracy using 5-fold cross validation? This should ensure that all parts of the data are being used evenly, preventing only easy examples from landing in the test set. Failing that, the authors would need to provide a much more robust explanation for why test accuracy is consistently higher.
> > > >
> > > > > 24. I am somewhat unconvinced by the authors' argument that the black-box will generalize better because it is trained on a larger number of examples. While one may expect the black-box to perform better (which it does, based on the accuracies for different levels of transparency), I do not believe it should be able to achieve higher accuracy on the test set. My understanding is that these datasets are large enough that there would be some overfitting on the training data. Furthermore, if the errorbars show standard deviation (and not standard error of the mean), then the difference could be statistically significant (once you divide by square-root of N), and is consistently present across all values of for the two plots mentioned.
> > > >
> > > > We shall address both points in tandem since they relate to the same issue that test performance often appears too good.
> > > > First and foremost, we did additional data analysis comparing the training and test accuracies of all hybrid models trained in Section 5.2.
> > > > - On COMPAS, 81% of the hybrid models have a lower test accuracy than training accuracy
> > > > - On Adult, 89% of the hybrid models have a lower test accuracy than training accuracy
> > > > - On ACS-Employ, 94% of the hybrid models have a lower test accuracy than training accuracy
> > > >
> > > > And in any cases where the test accuracy was better than training accuracy, the gap was never more than 0.5 ppts.
> > > > Therefore, we argue that better test set accuracies are not a “highly consistent” pattern.
> > > >
> > > > Then, how can this observation be consistent with Figures 10 and 11? Doesn’t Figure 10 c) suggests that COMPAS test accuracies are ~5 ppts larger than train? Doesn’t Figure 11 b) with 0.95 transparency suggests that test set accuracies can be 2 ppts larger? The answer to both questions is no: the key insight is that Figures 10 and 11 **do not show train/test accuracies** of the full hybrid model. Rather, they plot quantities related (but not equivalent) to the full accuracy.
> > > >
> > > > Figure 10 illustrates **Upper Bounds** on the train/test set accuracies (see Equation 15), which sum the errors of the prefix on its support and the unavoidable errors sent to the black-box due to data inconsistencies. Since these are Upper Bounds (not actual accuracies), the fact that they are higher on test is peculiar, but not indicative of any data leakage. In fact, we compared these Upper Bounds with the actual accuracies on the train/test sets:
> > > > - On average, on COMPAS, the upper bounds overestimate train accuracy by 1 ppts and test accuracy by 5 ppts
> > > > - On average, on Adult, the upper bounds overestimate the train accuracy by 2 ppts and the test accuracy by 4 ppts
> > > > - On average, on ACS-Employ, the upper bounds overestimate the train accuracy by 1.6 ppts and the test accuracy by 2.7 ppts
> > > >
> > > > **This explains why, in Figure 10, these upper bounds seem to be better on the test set, while the actual accuracy is in reality better on the train set.**
> > > >
> > > > Figure 11 illustrates the train/test accuracy of the black-box **evaluated on samples that land in $\bar{\Omega}$**. This quantity is not necessarily representative of the full hybrid model accuracy for two reasons.
> > > > The full performance must additionally account for the predictions made over $\Omega$.
> > > > The training error evaluates the black-box $h$ on training samples that land in $\bar{\Omega}$. However, this region $\bar{\Omega}$ was determined beforehand by minimizing Equation 15 on the same training samples. Seeing as Equation 15 penalizes the prefix for making errors on $\Omega$, it is possible that the hardest training samples (those that could not be predicted by a short rule list) were sent to $\bar{\Omega}$. Consequently, the distributions of training samples landing in $\bar{\Omega}$ is not necessarily the same as the distribution of fresh test samples landing in $\bar{\Omega}$.
> > > >
> > > > Note that all reported results are already evaluated using 5-folds cross-validation.

---

> > > > > ### Comment · Reviewer_ufqm · 2024-07-29
> > > > > **Thank you to the authors for the additional explanation**
> > > > >
> > > > > I thank the authors for clarifying the apparent discrepancy in their new explanation and I accept their reasoning. I would encourage them to add a version of this explanation to the paper (e.g., in the appendix).

---

> > > > > ### Author Response · Authors · 2024-07-29
> > > > > **Response to Reviewer ufqm (2/2)**
> > > > >
> > > > > > 1. I feel that the authors have not understood my issue in this point. Let's say that the dataset is divided into three parts D_train, D_val and D_test (at a 60-20-20 ratio) for training, validation and testing respectively. Then, in Section 5.2, for choosing $\alpha$, the authors have used an 80-20 split (of the full dataset, I assume) - so let us say they use D_train and D_val for training and D_test for testing. Finally $\alpha$ is chosen by looking at the accuracies on all these parts of the dataset. In other words, the estimate $\hat{\alpha}$ that the authors arrive at depends on D_train, D_val and D_test.
> > > > > Now, in Section 5.3, when they fit the model on just D_train, they are using $\hat{\alpha}$(D_train,D_val,D_test) during fitting, so in other words, they are indirectly using information from D_test to train the model. The authors can address this in one of three ways: (1) reproduce the figures in Section 5.2 using only D_train and D_val (i.e., not using D_test while choosing $\alpha$); (2) cross-validate to determine $\alpha$ along with other hyper-parameters; or (3) acknowledge this issue in the paper itself state that they do not expect the results to be significantly different because their method for choosing $\alpha$ is fairly heuristic.
> > > > >
> > > > > We understand your point and agree to your third solution: (3) acknowledge in the paper that our method for choosing a value of $\alpha$ is simply a rule of thumb, and does not result from a thorough optimization method - although doing so could only improve our results. Indeed, the purpose of Section 5.2 was to investigate the effectiveness of an instantiation of the Pre-Black-Box paradigm (with HybridCORELSPre) to learn hybrid interpretable models, and more specifically the role of the regularization coefficient $\alpha$. Our main takeaway is that a moderate value of this coefficient (such as 1 or 2) leads to some improvement most of the time, and we arbitrarily set it to 1 for the remainder of the paper. Again, this choice does not result from an optimization procedure, and the exact best value of $\alpha$ would depend on the experiment at hand - and in particular on the chosen transparency level, on the type of black-box and on the used dataset.
> > > > >
> > > > >
> > > > > > 20. Firstly, in accordance with point 1 above, let us distinguish between test and validation accuracy, because $\alpha$ is a hyperparameter and should be chosen based on training and validation data (not test data). Now, I understand that the "improvement rate" is the percentage of experiments in which you get higher validation accuracy, but I am trying to understand why this is a better metric than the average validation accuracy across experiments. Is it because there is very high variance? Can the authors provide a plot of the improvement rate as a function of $\alpha$ ?
> > > > > Furthermore, if the improvement rate is highest for $\alpha = 1$, why have the authors chosen to use $\alpha = 2$ in Section 5.3? This feels very arbitrary, and reduces reader confidence in the whole analysis.
> > > > > Finally, is there any ML literature that could recommend or support using a quantity like improvement rate to choose hyper-parameters in hybrid interpretable models, or is this an innovation presented in this paper?
> > > > >
> > > > > We chose to focus on the improvement rate as it reflects how a particular value of $\alpha$  is able to consistently improve the black-box performances over a large set of scenarios. Indeed, the purpose of Section 5.2 was not to fine-tune HybridCORELSPre, but to investigate how the Pre-Black-Box paradigm works in practice - and in particular its key parameter $\alpha$, for different configurations.
> > > > > Indeed, in practice, one would need to perform hyper-parameter optimization for $\alpha$, for instance by picking the value of $\alpha$ leading to the highest validation accuracy.
> > > > > For a single experiment, the notion of improvement rate would not even make sense - note that we used it here because our objective was to investigate the Pre-Black-Box paradigm but not for the sake of optimizing its value.
> > > > >
> > > > > For Section 5.2, methodologically speaking, working with an average accuracy improvement (rather than an improvement rate) would have several issues: since raw accuracies vary between datasets and black-box, a particular setup could completely hide the tendencies over the other ones. Despite these concerns, in practice, working with an average accuracy improvement would have led to the same results - as reported in the following table (for experiments with the Adaboost black-box, which are those mentioned in the paper):
> > > > >
> > > > > | $\alpha$ | Improvement rate (\%) | Average absolute improvement |
> > > > > |:--------:|:---------------------:|:----------------------------:|
> > > > > | 	1	|      	80.0     	|        	0.0048        	|
> > > > > | 	2	|     	93.33     	|        	0.0052        	|
> > > > > | 	4	|     	93.33     	|        	0.0048        	|
> > > > > | 	6	|      	80.0     	|        	0.0037        	|
> > > > > | 	8	|     	66.66     	|        	0.0027        	|

---

### Author Response · Authors · 2024-07-03
**We thank the reviewers for their constructive comments and feedback**

We want to express our sincere thanks for the detailed reviews of our work and the overall positive comments and feedback.
We have answered each review individually, providing additional clarifications regarding important questions or limitations. Furthermore, we have uploaded the revised version of our paper, in which all modifications appear in blue.

We remain available to answer any remaining questions. Once again, we thank you for your consideration and all the time invested in this review process.

---

### Author Response · Authors · 2024-07-31
**Thank you for your detailed feedback**

We have updated the paper following the last comments of Reviewer ufqm, and hope our modifications and clarifications address all the remaining concerns. The two main modifications (colored blue) are the following:
- Section 5.2 has been updated. Equations 16 and 17 were added to clarify the quantities reported in the y-axis of Figures 10 and 11. Moreover, the section justifies the use of improvement rate instead of average accuracy, as well as the fact that setting $\alpha=1$ is a "rule of thumb’’. Future work should investigate cross-validation as a means to tune this hyperparameter.
- The Appendix D was added to discuss the concerns of Reviewer ufqm regarding test performances that are better than training set in Figures 10 and 11. We lay these concerns to rest by reporting the train/test accuracies of the full Hybrid Interpretable Models. We also explain why the performances reported in Figures 10 and 11 are better on test and train.

Once again, we thank all reviewers for their thorough reviews and remain available for any additional comments or questions.

---

### Decision · Action_Editor_xF5u · 2024-07-29

**Recommendation:** Accept with minor revision

**Comment:**

One detailed reviewer-author discussion reveals an interesting discrepancy between training/testing performance, as well as some unresolved questions about the tuning of $\alpha$. The authors did not completely resolve these issues, and I strongly encourage the authors to consider the reviewer as being representative of an interested reader, who might have similar questions while reading the final manuscript. I encourage the authors to make changes to the manuscript to resolve these issues to the best of their ability, with minor changes.

**Audience:**

All reviewers agree that the paper is suitable for the TMLR audience. I strongly agree.

**Claims And Evidence:**

All reviewers agree that claims and evidence are supported.

The authors have responded to reviewer comments and updated their manuscript regarding some minor claims and evidence / presentation issues, such as realizability assumption, tuning of hyperparameter $\alpha$, extra references, etc. as well as minor typos.

---

> ### Author Response · Authors · 2024-08-03
> **Camera Ready**
>
> We have uploaded the camera ready version of our paper. It includes detailed discussions regarding the discrepancy between training and test performance (and a new dedicated appendix section) and regarding the tuning of $\alpha$.
>
> Once again, we would like to thank you for this insightful and thorough review process.